



# 1 Successional patterns of (trace) metals and microorganisms in the Rainbow

# 2 hydrothermal vent plume at the Mid-Atlantic Ridge

Sabine Haalboom[1,*], David M. Price[1,*,#], Furu Mienis[1], Judith D.L van Bleijswijk[1], Henko C. de
Stigter[1], Harry J. Witte[1], Gert-Jan Reichart[1,2], Gerard C.A. Duineveld[1]
[1] NIOZ Royal Netherlands Institute for Sea Research, department of Ocean Systems, and Utrecht University, PO Box 59,
1790 AB Den Burg, Texel, The Netherlands
[2] Utrecht University, Faculty of Geosciences, 3584 CD Utrecht, The Netherlands
[*] These authors contributed equally to this work
[#] Current address: University of Southampton, Waterfront Campus, European Way, Southampton, UK,
SO14 3ZH.
sabine.haalboom@nioz.nl; D.M.Price@soton.ac.uk
**Keywords:** Rainbow vent; Epsilonproteobacteria; Hydrothermal vent plume; Deep-sea mining; Rare
earth elements; Seafloor massive sulfides

## Abstract

Hydrothermal vent fields found at mid-ocean ridges emit hydrothermal fluids which disperse as neutrally
buoyant plumes. From these fluids seafloor massive sulfides (SMS) deposits are formed which are being
explored as possible new mining sites for (trace) metals and rare earth elements (REE). It has been
suggested that during mining activities large amounts of suspended matter will appear in the water column
due to excavation processes, and due to discharge of mining waste from the surface vessel. Understanding
how natural hydrothermal plumes evolve as they spread away from their source and how they affect their
surrounding environment may provide some analogies for the behaviour of the dilute distal part of
chemically enriched mining plumes.



This study on the extensive Rainbow hydrothermal plume, observed up to 25 km downstream from the
vent site, enabled us to investigate how microbial communities change in the presence of a natural plume.
The (trace) metal and REE content of suspended particulate matter (SPM) was determined using HR-ICP
mass spectrometry and the microbial communities of the neutrally buoyant plume, above plume-, below
plume-, and near-bottom water and sediment were characterised by using 16S rRNA amplicon sequencing
methods. Both vertically in the water column and horizontally along the neutrally buoyant plume,
geochemical and biological changes were evident as the neutrally buoyant plume stood out by its
enrichments in (trace) metals and REEs, of which the concentrations changed as the plume aged. This
was also reflected in the background pelagic system as Epsilonproteobacteria started to dominate and the
biodiversity appeared to reduce with distance away from the Rainbow hydrothermal vent field. The
Rainbow hydrothermal plume provides a geochemically enriched natural environment, which is a
heterogeneous, dynamic habitat that is conducive to ecological changes in a short time span.

## 1     Introduction

Hydrothermal vent fields found at mid-ocean ridges and back-arc basins are known for discharging fluids
rich in potential microbial energy sources such as $H_2$, $H_2S$, $CH_4$, $NH_4$ and Fe (Jannasch and Mottl, 1985;
McCollom, 2000). In addition, they are characterised by the presence of polymetallic sulfide deposits
containing high grades of metals like Cu, Co, Zn and rare earth elements (REE). Because of the steadily
increasing demand for these metals, and their geo-political distribution on land, hydrothermal vent
deposits are explored as new possible mining sites (Hoagland, 2010). Since such areas accommodate
unique and vulnerable marine life, serious concerns exist about the environmental sustainability of
seafloor massive sulfide (SMS) deposit mining (Boschen et al., 2013; Collins et al., 2013), especially with
regards to the effects of plumes generated during the excavation of ores and by the return flow of wastes
in the vicinity of hydrothermal vents (Ramirez-Llodra et al., 2011; Vare et al., 2018). As SMS mining
will concentrate on deposits around hydrothermal vents, and not on active vents or chimneys due to
technical risks associated with high temperatures (Gwyther et al., 2008), it is likely that the background
and extinct vent communities (from microorganisms to megafauna) will be impacted through habitat loss,



mechanical destruction, noise, smothering and bioaccumulation of toxic substances (Levin et al., 2016). However, knowledge about the background ecosystem and natural plume is sparse, as the vents and their proximal fauna have attracted most of the attention, for example in microbiology (e.g. Han et al., 2018; Cerqueira et al., 2018).

To fill this gap, the Dutch TREASURE project (STW-NWO) was focussed on describing the structure of the background pelagic and benthic communities of an active hydrothermal vent site with SMS deposits on the Mid-Atlantic Ridge (MAR). The Rainbow hydrothermal vent south of the Azores was selected for this study as it ejects one of the most prominent and persistent natural plumes on the MAR. Basic knowledge of natural plumes is essential to assess mining impacts for two reasons: firstly, hydrothermal plumes represent a distinct ecosystem in itself, which under the influence of currents may extend tens of kilometres away from its point of origin. The same currents will also disperse mining plumes, created in the vicinity of the hydrothermal vent. These mining plumes are therefore likely to interfere with the hydrothermal plume and thus potentially alter baseline (T0) conditions. Secondly, understanding natural plume processes may reveal how ecosystems adapt to elevated turbidity and co-occurring changes in the chemical environment.

Since the discovery of the Rainbow hydrothermal vent field in 1996 by German et al., several relationships concerning the composition of the hydrothermal fluid and the associated sediment formed by precipitation from the hydrothermal plume have been established. For example it was shown that the underlying host rock influences the hydrothermal fluid composition (Wetzel and Shock, 2000). Geochemical investigation of sediment by Cave et al. (2002) at distances of 2 to 25 km from the Rainbow hydrothermal vent field showed enrichments of Fe, Cu, Mn, V, As and P as a result of fallout from the hydrothermal plume. Studies of other hydrothermal vent systems demonstrated that deposition from the plume is partially being influenced by microbial activity which enhances scavenging and oxidation rates of metals (e.g. Cowen and Bruland, 1985; Cowen et al., 1990; Mandernack and Tebo, 1993; Dick et al., 2009), with implications for the local ocean geochemistry.

Microbial activity within the plume is fuelled by redox reactions that provide energy for chemotrophic microbial taxa. The abundance of energy sources within plumes support a plethora of chemotrophic



microbial communities (e.g. Frank et al., 2013; Anantharaman et al., 2015). Plume microbial communities
can be distinct or relatively similar to background communities (Dick and Tebo et al., 2010; Sheik et al.,
2015; Olins et al., 2017), with plume associated bacteria originating from either seafloor communities,
background seawater communities or from growth within the plume (Dick et al., 2013). Djurhuus et al.
(2017) observed the dilution of vent associated microorganisms with increased redox potential,
suggesting that communities associated with the rising plume would disperse with distance from the vent
on a scale of metres, showcasing a variable community within the plume. After its initial rise, a
hydrothermal vent plume becomes neutrally buoyant and is dispersed over potentially hundreds of
kilometres (German and Sparks, 1993; Dymond and Roth, 1988), however this portion of the plume has
not been sampled in a similar manner to identify microbial community patterns.
Overall, little is known about the chemical fractionation or microbial assemblages within the neutrally
buoyant plume as it ages and disperses from the hydrothermal vent field. Notably, due to the lack of
quantified characteristics of SMS mining plumes (especially the discharge plume), the T0 influence of
this hydrothermal plume may act as an analogue for future mining plume impacts. Although it should be
kept in mind that discharge plumes will have different physical characteristics as these plumes will have
a higher initial density and therefore would tend to sink rather than maintain buoyancy and may have a
different release depth. However, the natural plume could serve as an analogue for the finest and slowest
sinking fraction of suspended solids in the mining plume. In this study, water column and sediment
samples from the Rainbow hydrothermal vent area were investigated. Geochemical and biological
changes were tracked vertically in the water column and horizontally along the neutrally buoyant plume,
to study the heterogeneity in the background pelagic system that was influenced by the hydrothermal
plume. By utilising a range of methods that could be useful as monitoring techniques and describing
background environments that may be influenced by SMS mining, we contribute to site specific
knowledge of the Rainbow hydrothermal vent plume behaviour, associated (trace) metal enrichments and
microbial community composition.






## 2 Material and methods

### 2.1 Study site

The Rainbow hydrothermal vent field (Fig. 1) is located on the Mid Atlantic Ridge (MAR) at 36°13.80 N, 33°54.14 W at approximately 2300 m water depth, southwest of the Azores. The vent field is located on the western flank on the non-volcanic Rainbow Ridge, in an offset between the South Alvin Mid Atlantic Ridge (AMAR) and AMAR segments of the MAR (Fouquet et al., 1998; Douville et al., 2002). It is located at the intersection between the non-transform fault system and the ridge faults (Charlou et al., 2002), making this vent field tectonically controlled. The vent field, which is approximately 100 by 250 m in size, is underlain by a basement composed of ultramafic rocks (Edmonds and German, 2004). The ultramafic setting of Rainbow is atypical for the region, which is dominated by basalt hosted vent systems (Douville et al., 2002). Due to serpentinization reactions during the circulation of the hydrothermal fluid in the peridotite basement rocks, the Rainbow vent field produced plumes particularly enriched in transition metals (notably Fe, Mn and Cu) and REE (Douville et al., 2002; Findlay et al., 2015). On the contrary the plumes are depleted in hydrogen sulfides (Charlou et al., 1997; Douville et al., 2002), resulting in relatively high metal/sulfide ratios. Consequently, the chimneys and the SMS deposits of the Rainbow hydrothermal field are enriched in Cu, Zn, Co and Ni when compared to vent systems with a basaltic host rock (Charlou et al., 1997).

The vent field consists of 10 active, high temperature (365 °C) black smokers and emits an extensive plume with a distinct chemical composition compared to the ambient seawater (Severmann et al., 2004). It is considered the largest and widest spreading plume in the region (German et al., 1996), rising up to 200 m above its source and was traced over 50 kilometres (Severmann et al., 2004). The plume dispersion is controlled by the local hydrodynamic regime and topography (Thurhnerr and Richards, 2001; Thurnherr et al., 2002), moving predominantly to the north and east around the Rainbow Ridge with an average current speed of 5-6 cms$^{-1}$ and continues in a northward direction along the southern and eastern side of the rift valley of the AMAR segments (Edmonds and German, 2004). The plume characteristics and prior knowledge of its behaviour make the Rainbow vent field a suitable site to study neutrally buoyant plumes.





## 2.2 Water column profiling and sampling

Water samples and sediment cores were collected along the gradient of the plume during RV *Pelagia* cruise 64PE398 in April 2015. Five putatively distinct biotopes were sampled: (i) above plume (1000 m water depth), (ii) plume, (iii) below plume (10 metres above bottom), (iv) near-bottom water and (v) sediment. Using CTD casts with a CTD-Rosette system, the plume was traced in real time using turbidity as an indicator, measured in NTU with a WETLabs turbidity sensor. Other variables measured included temperature (°C), salinity (PSU), density ($\sigma$-$\theta$, kgm$^{-3}$), dissolved oxygen (mlL$^{-1}$) and chlorophyll ($\mu$gL$^{-1}$). At five stations, continuous yoyo CTD-casts were taken over the course of 12 hours, to study the temporal changes of the hydrothermal plume.

A total of 41 water samples were collected using 12 L Niskin bottles from eleven downstream stations, two distal downstream stations and three upstream stations. Once the CTD was back on deck, three distinct water samples were immediately taken for suspended particulate matter (SPM), trace metals, and the microbial community. Additional intermittent water samples were taken for nutrients and suspended particulate organic matter (Table 1).

Sediment and near-bottom water samples were collected with a NIOZ designed box corer of 50 cm diameter equipped with a top valve to prevent flushing, subsequently trapping more than 1.5 litres of near-bottom water (van Bleijswijk et al., 2015). In total eight cores were collected (Table 1). Due to unsuitable coring substrates, CTD locations and coring sites did not always follow the same track. Cores were taken on the eastern part of the Rainbow Ridge, continuing in the basin east of the ridge, while two cores were taken on the north-western flank of the ridge, following the path of the plume.

## 2.3 Suspended particulate matter analysis

From each 12 L Niskin bottle, two 5 L subsamples were collected to determine the concentration of SPM. The subsamples were filtered on board over pre-weighed 0.4 µm polycarbonate filters. The filters were rinsed with ~10 ml of Milli-Q water to remove salt, while still applying under pressure, and subsequently stored at -20 °C on board. Prior to analysis the filters were freeze dried. The samples were weighed in



duplo, or once again if the difference between the two measurements was 0.03 mg or more. To yield SPM
concentrations, the net dry weight of the SPM collected on the filters, corrected by the average weight
change of all blank filters, was divided by the volume of filtered seawater. Subsequently, the filters were
examined using a Hitachi TM3000 table-top SEM connected to an EDS-detector to visualize content of
the SPM and analyse the chemical composition.

**2.4    Chemical analysis**
In order to examine the trace metals present in and around the hydrothermal plume, water samples were
filtered over acid-cleaned 0.45 µm polysulfone filters directly from the Niskin bottle at ambient
temperature while applying under pressure. A water barrel in between the filtration holder and pump
allowed for volume measurements of water filtered. The filters were subsequently stored at -20 °C until
further examination and were left to dry in an Interflow laminar flow bench at room temperature prior to
analysis. The filters were placed in acid-cleaned Teflon vials and were subjected to a total digestion
method. For this purpose a mixture of 6.5 ml $HNO_3$ (ultrapure)/HF (10:1) solution, 1 ml HCl and 1 ml
$HClO_4$ was added to the vials, after which the vials were covered and placed in an Analab hotblock for
48 hours at 125 °C. After the filters were completely dissolved, the covers were taken off from the vials
and the vials were left for 24 hours in order to evaporate the acids. Finally, the residue was taken up again
in 10 ml $HNO_3$, pre-spiked with 5 ppb scandium and 5 ppb rhodium as internal standards. A HR-ICP-MS
(Thermo Element II) was used to analyse the concentrations of major- and trace metals, as well as REEs.

**2.5    Microbial community**
Three distinct samples of 2 L of water were collected from three different Niskin bottles for Next
Generation Sequencing (NGS). The water was filtered immediately after collection through a 0.2 µm
polycarbonate filter (Nuclepore) facilitated by a vacuum of 0.2 bar at 4 °C, to limit DNA degradation.
With a sterilised spatula, >0.25 grams of surface sediment were scraped off from the box cores whilst 1.5



litre of overlying (near-bottom) water was filtered as above. Filters were stored in a 2 ml cryo-vial and all
samples were stored at -80 °C on board.
DNA was extracted using a Power Soil DNA Isolation Kit (MoBio, now Qiagen) according to the
manufacturer's protocol. Each DNA extract concentration was quantified using a Qubit 3.0 fluorimeter
(Qiagen, Inc.) and stored at -20 °C before amplification. Extracts were combined with Phusion Taq
(Thermo Scientific), High Fidelity Phusion polymerase buffer and universal primers to amplify the V4
region of 16 S rDNA of bacteria and archaea (Table 2), with unique molecular identifier (MID)
combinations to identify the different samples. All negative controls from all PCR series were labelled
with the same unique MID.  The PCR settings were as follows: 30s at 98 °C, 29 cycles (10s at 98 °C, 20s
at 53 °C, 30s at 72 °C) and 7 minutes at 72 °C. Four and three samples were re-run at 30 and 32 cycles,
respectively, in order to yield enough product. Each sample was subjected to the polymerase chain
reaction (PCR) protocol in triplicate and processed independently to avoid bias. 5 µl of product was used
to screen the products on an agarose gel. The remaining 25 µl of each triplicate was pooled to evenly
distribute the DNA, split into two slots and run on a 2 % agarose gel at 75 volts for 50 minutes. Sybergold
stain was applied post run for 20-30 minutes before cutting the 380 bp bands out with a sterilised scalpel
over a blue light to avoid UV damage. The two bands of mixed triplicates were pooled, purified using the
Qiaquick Gel Extraction Kit (Qiagen, Inc.) and quantified with a Qubit™ 3.0 fluorometer (Qiagen, Inc.).
Samples were pooled in equimolar quantities together with blank PCR controls. The pooled sample was
concentrated using MinElute™ PCR Purification columns (Qiagen Inc.) as described by the manufacturer
and sent to Macrogen (South Korea) for sequencing. Sequencing was undertaken with a Roches GS FLX
instrument using Titanium chemistry on a one-eight region gasket and Roche GS FLX instruments.
Sequence processing was undertaken as described by van Bleijswijk et al. (2015), using a QIIME pipeline.
Sequences shorter than 250 bases and average Q scores below 25 were removed. The OTU sequences
(>98 % similarity) were classified (>93 % similarity) based on a recent SILVA SSU database (release
132; Yilmaz et al. 2014). Single reads were excluded and all data were standardised to remove any
disproportionate sampling bias.





## 2.6    Statistics

Unconstrained ordination techniques were utilised to distinguish biotopes and general community patterns. Non-metric Multi-Dimensional Scaling plots (NMDS) were created based upon Bray-Curtis similarity matrices of square root transformed microbial community assemblages. Group average clustering was also utilised in order to quantify similarities between the samples. ANalysis Of SIMilarities (ANOSIM) was subsequently used to statistically test community distinctions based upon presumed biotopes (sediment, near-bottom water, below plume water, plume water and above plume water). In addition, all water column samples were plotted in separate NMDS plots to observed patterns in greater detail. Physical properties of all water samples (station, depth, turbidity and location) were depicted in a NMDS plot to observe sample similarities. These environmental data were normalised and Euclidean distance was used to create a similarity matrix. The relationship between Fe and turbidity was tested with a linear regression analysis. Trace metals and REE were normalised to Fe, since it is the primary particle-forming element at all stages of plume dispersion, giving insight in the chemical behaviour. All multivariate statistics were undertaken in Primer™ V6 (Clarke and Gorley, 2006).

Shannon-Wiener index (log e) was calculated as a diversity measure. Biodiversity differences between biotopes were tested with the non-parametric test Kruskal-Wallis with pairwise comparisons as the data did not meet normality or homogeneity assumptions, even after transformation. These statistical tests were undertaken in SPSS.

A SIMililarities PERcentage analysis (SIMPER in Primer v6) was applied on the microbial class level with a cut off for low contributions at 90 % based on Bray-Curtis similarity matrix to characterise the community composition based on groups contributing to intra biotope similarities. Relationships between environmental variables and microbial classes as a percentage of each composition within the plume, were tested with Pearson correlation and hierarchical clustering to identify broad response groups.



## 3 Results

### 3.1 Water column characteristics

Temperature, salinity and density plots indicated that the water column at each location had similar physical traits, whereby three different water masses could be distinguished (Supplement Fig. S1). The surface Eastern North Atlantic Central Water (ENACW) was characterised by a temperature, salinity and density at the surface of 18 °C, 36.4 PSU and 26.2 kgm$^{-3}$ to 11 °C, 35.5 PSU and 27.2 kgm$^{-3}$ at the bottom of the water mass. The underlying Mediterranean Outflow Water (MOW) was characterised by a temperature of 7.5-11 °C, a salinity of 35.4-35.5 PSU and a density of 27.2-27.75 kgm$^{-3}$. The North Atlantic Deep Water (NADW) was characterised by temperatures ranging from 4 to 7.5 °C, salinity of 35.0 to 35.4 PSU and a density of 27.75 to 27.825 kgm$^{-3}$ (Emery and Meincke, 1986). The neutrally buoyant plume was centred around the 27.82 kgm$^{-3}$ isopycnal, as illustrated in Figures 2 and 3.

### 3.2 Turbidity and plume dispersion

Against a background of non-plume influenced waters with typical concentrations of SPM of 0.04 mgL$^{-1}$ (0.015 NTU), the neutrally buoyant plume stands out as a layer of distinctly higher turbidity values (i.e. higher SPM concentrations) in the depth interval of 1750 – 2400 m (Fig. 2). The apparent continuity of this turbid water layer, especially to the NE of the Rainbow field, and lack of similarly turbid waters in the bottom waters below the plume, link the plume to Rainbow and preclude an origin in local sediment resuspension.

At downstream stations, a consistent trend of decreasing turbidity and increasing vertical dispersion was noted. At station 27, 3.5 km north of Rainbow, maximum turbidity in the core of the plume was 0.15 NTU (0.09 mgL$^{-1}$) and plume thickness was about 105 m, whilst at station 46, 15.2 km east of Rainbow, maximum turbidity was only 0.08 NTU (0.06 mgL$^{-1}$) and plume thickness was 275 m. Away from the main plume path, station 47 and 49 (13.8 and 16.5 km from Rainbow, respectively) showed a diluted signature similar to that observed at the most distal stations along the main plume path. Despite being most proximal to Rainbow, station 16, located 1.0 km downstream of Rainbow, showed a relative low



turbidity of 0.015 NTU (0.04 mgL$^{-1}$). Since the plume is more constrained closer to the source, the main
body of the narrower plume could have been missed with the CTD. Stations upstream of the vent site
(station 13 and 28, 4.2 and 7.5 km southwest of Rainbow respectively and station 40, 3.6 southeast of
Rainbow) displayed low turbidity values, ranging between 0.01 and 0.02 NTU (0.04 mgL$^{-1}$) (Fig. S2).
The CTD profiles from stations 42 and 49 (4.9 and 16.5 km north of Rainbow respectively) both displayed
highest turbidity in the lower hundreds of metres above the seafloor, with instances of seafloor contact
during time of sampling. Therefore no samples could be taken below the plume at these stations. The
assumption that the plume is subject to vertical movement is supported by observations made during 12-
hour CTD yoyo casts carried out at station 27 (Fig. 3). Along with vertical displacements of the 27.82
kgm$^{-3}$ isopycnal on the order of 150 m, likely reflecting internal tidal motions, the hydrothermal plume
was found to also move up and down, at times touching the seafloor.

**3.3      Enrichment of (trace) metals compared to the ambient seawater**
NMDS ordination (Fig. 4) based on Euclidean distance resemblance of normalised element/Fe molar ratio
data of all collected water samples (2D stress = 0.03), revealed a clear distinction of the different samples.
Most outstanding are the samples from above plume waters, indicating that the chemical composition is
different from the other samples.
The remaining samples showed less variation, nonetheless the samples collected from below the plume
and the samples collected away from the main path of the plume can be distinguished. This shows that
the hydrothermal plume can be characterised by its chemical composition. When comparing samples
taken in the turbidity maximum of the plume to the above plume water samples taken at 1000 m water
depth it is found that Fe, Cu, P, V and Pb are enriched by factors of ~80, ~90, ~17, ~52 and ~25
respectively. Elements with a more moderate degree of enrichment are Co, Mn, Zn, Al and Ni, with
enrichment factors of ~8.0, ~2.5, ~10.3, ~1.4 and ~1.6, respectively. The REEs were enriched by a factor
of 5 to 40 relative to the clear water. U, Ti and Ca are slightly enriched at turbidity maxima, by factors of
~1.3, ~1.6 and ~1.2, respectively. In and Sn are depleted compared to the clear water above the plume.





## 3.4    Geochemical gradients within the hydrothermal plume

Within the hydrothermal plume, geochemical evolution is found as the plume disperses. Visual examination of the samples with the SEM coupled with chemical analysis performed with the EDS-detector revealed that the SPM within the plume close to the Rainbow hydrothermal vent at station 32 (2.9 km north of Rainbow) mainly consisted of Fe-sulfides. In the plume samples further downstream, Fe is mainly present as Fe-oxides, Fe-hydroxides or bound in alumino-silicates.

Chemical examination of the samples showed gradients in the element/Fe molar ratios along the path of the plume as well as off the main path of the plume at upstream and the most distal downstream stations. Since the Fe concentration is linearly related to the turbidity (Fig. 5) ($R^2$ = 0.9356), normalisation to Fe reveals relative enrichments or depletion of common elements. The chalcophile elements Co, Cu and Zn show a partly-linear relation steepening with increasing Fe concentration (Fig. 6A), indicating that the element/Fe molar ratios are elevated close to the source but decrease towards the more distal sites (Fig. 7A). One exception is the Zn/Fe molar ratio, which is elevated at station 37, 39 and 44. The oxyanions P and V are linearly related to Fe (Fig. 6B), therefore they also display more or less constant molar ratios, both upstream and downstream of Rainbow (Fig. 7B). The REE show a partly-linear relation levelling-off with increasing iron concentrations (Fig. 6C). Within the plume this is displayed as increasing element/Fe molar ratios towards station 44, with station 42 as an exception, followed by a constant or slightly decreasing molar ratio from station 44 onwards (Fig. 7C). The Ca/Fe molar ratios ranged between 0 and 15 for most of the downstream stations, apart from the stations further downstream (47 and 49), which displayed slightly higher Ca/Fe molar ratios. Upstream station 28 had a Ca/Fe molar ratio similar to those found at station 47 and 49 and upstream station 40 was found to have a significantly higher Ca/Fe molar ratio. Other analysed elements, Mn, Al, Ni, In, Pb, Ti and U showed no clear relationship with the Fe concentration (Fig. 6D). However, within the plume it was found that the Mn/Fe molar ratio is lower than at the upstream stations or the more distal downstream stations.



### 3.5 Microbial assemblages in water column biotopes

Samples from sediment, near-bottom water and above plume water contained microbial communities which clustered distinctly from each other and from plume and below-plume communities (Fig. 8). In particular, sediment and near-bottom water samples have communities that are very dissimilar from the overlying water column samples. Sediment samples appeared to cluster in a straight line suggesting some sort of gradient of similarity along the ordination axis, though no apparent patterns were observed when independently plotted. The near-bottom water samples were relatively dispersed in the MDS plot suggesting a more variable community. Samples taken at the upstream station 13 from below-plume and plume depths showed no similarity with analogous samples from the other stations, except for the above plume community which is consistent with other stations. In general, plume and below-plume communities were more similar nearer to the vent source, with stations further downstream displaying greater dissimilarity (Fig. 9, Fig. S3).

Group average cluster analysis showed high level of dissimilarity, i.e. large community variation, between and within biotopes. ANOSIM revealed all putative biotopes that were sampled had distinct communities (Global R = 0.738; p = 0.001; 999 permutations), except for plume and below plume samples which could not be distinguished statistically (Global R = -0.091; P = 0.861). The two seemingly unique samples from station 13 also tested significantly distinct, but with a low number of permutations (<999) due to low replication (n=2).

### 3.6 Univariate biodiversity

Plume and below plume samples were less diverse than sediment samples, whilst diversity in the plume was lower than in near-bottom water samples (Kruskal-Wallis: $\chi^2$ (4) = 36.127, P <0.01). In general, plume diversity was low (Fig. 10), but further differences were not statistically significant, likely due to limited replication and intra biotope variation.

The plume microbial community at sites upstream of Rainbow and at the immediate downstream sites (stations 28, 16 and 27) showed similar and relatively high biodiversity (>4.5) (Fig 11). Plume



biodiversity at the sites further away from Rainbow gradually decreased until station 46, which displayed the lowest Shannon index value of 2.4. Distant stations 47 and 49, showed biodiversity rising to a more moderate index value around 3.5.

### 3.7    Species composition

Results of the SIMPER analyses showing the contributions of taxa composition to similarities within biotopes (Table 3), mirrored the NMDS and ANOSIM results whereby the similarity of community composition in each biotope was dominated by a different makeup of the microbial community. The Archaeal class Nitrososphaeria (Marine group 1 archaea) contributed the most to similarity within the above and below plume water communities, while also being very common in all water samples. Alphaproteobacteria, Gammaproteobacteria and Deltaproteobacteria also constituted as a large makeup of all biotopes in the area. The class Epsilonproteobacteria were largely absent from above plume samples being not influenced by the plume, and only contributed <2 % to near-bottom water communities. By contrast, Epsilonproteobacteria were dominant in plume water samples (accounting for >35 % of the community), and were the fifth most dominant taxon in below plume water samples contributing 8.9 % of the community.

Epsilonproteobacteria accounted for about 20 % of the plume community at stations near the vent. Beyond the near vent stations, an increase in relative abundance of Epsilonproteobacteria with distance from vent was observed, accounting for 64 % of the community at the distant station 46 (Fig. 12). Alphaproteobacteria, Deltaproteobacteria and Gammaproteobacteria appeared to become less dominant with distance from the plume source (Fig. 12). The communities at distant stations 47 and 49 were less dominated by Epsilonproteobacteria (around 40 %). Below plume communities were dominated mostly by Nitrososphaeria (Marine group 1 Archaea) whereby Nitrosphaeria became more dominant with distance from the plume source likewise as the Epsilonproteobacteria in the plume. Correlations between environmental variables (elemental chemistry and physical properties) and all microbial classes observed in the plume were evident and appeared class specific (Fig. S4). The hierarchical clustering revealed eight broad response groups, which displayed different relationships with the environmental variables.



## 4 Discussion

Using a multidisciplinary approach in which physical, geochemical and ecological data were collected from the Rainbow vent neutrally buoyant plume and its underlying sediment, we aimed to expand knowledge of the T0 state of the background ecosystem of a hydrothermal vent. Such knowledge is deemed essential to be able to assess (potential) impacts of future deep-sea SMS mining. We found differences between the plume and background water composition with identified distinct biotopes. In addition, pertinent chemical and biological gradients within the extensive Rainbow hydrothermal vent plume were evident.

### 4.1 Physical constraints of plume location and behaviour

The plume was observed within the NADW mass, constrained to an isopycnal density envelope of 27.82 $kgm^{-3}$ (Fig. 2 and 3). Using turbidity measurements and presumed plume path, we traced the plume up to 25 km away from the vent source, in agreement with observations made by German et al. (1998) of a plume greater than 50 km, that is controlled by local hydrodynamics and topography. Unexpectedly, in the basin upstream of the Rainbow vent field a turbidity peak at 1975 m water depth resembling a plume was observed as well, confounding our assumption of a clear water column at upstream stations and distant downstream stations. This indicates that the plume is reaching further than previously observed by Thurnherr and Richards (2001) and German et al. (1998). This is exemplified by the local variation in microbial community composition of upstream stations (Fig. 12) and is supported by the relatively low Ca/Fe molar ratio at station 28 (Fig. 7), indicating hydrothermal influence. In addition, the observed variability of plume strength and vertical position (Fig. 3) indicate that local fluctuation in the current regime and tidal motions influence the plumes behaviour. This dynamic behaviour has implications for surveys designs and should be considered when monitoring natural and man-made plumes, such as mining-related plumes. Prior insight into plume extension and behaviour is required for the identification of adequate control sites and for tracking of plume evolution in future impact studies.





## 4.2 Plumes influence on the water column chemical and microbial make-up

The neutrally buoyant plume introduced pelagic heterogeneity in terms of chemical and microbial composition, which is supported by the vertical classification of the different biotopes. The neutrally buoyant plume was evidently enriched in metals and REE compared to overlying clear water. Element concentrations were found to be in line with those found by German et al. (1991) and Edmonds and German (2004) who have studied the Trans-Atlantic Geotraverse (TAG) hydrothermal plume and the Rainbow hydrothermal plume, respectively. Our chemical results from Rainbow also match with those of Ludford et al. (1965), who have studied vent fluid samples from the TAG, Mid-Atlantic Ridge at Kane (MARK), Lucky Strike and Broken Spur vent sites, i.e. element concentrations were found to be in the same order of magnitude.

The distinctive chemical composition of the plume samples (e.g. metal concentrations) affects chemotrophic microbial growth within the plume as indicated by the typical microbial community in plume samples. Unlike Sheik et al. (2015), we observed a clear and consistent separation between communities in the plume and those in above-plume samples. The influence of MOW on the above-plume community could also play a role, as oceanic water masses can harbour different microbial communities (Agogue et al., 2011). However, the palpable presence of a plume in the turbidity data with supporting chemical measurements, and the occurrence of vent associated Epsilonproteobacteria (Olins et al., 2017; Djurhuus et al., 2017) and other vent associated groups such as SUP05 (Sunamura et al., 2004), point to a unique chemical environment. Here chemosynthetic communities flourish and give rise to independent biotopes in the neutrally buoyant plume kilometres downstream of the vent site.

Below-plume communities were not distinct from the plume biotope, although instead of Epsilonproteobacteria, the ubiquitous class Nitrososphaeria was the most dominant group, reflecting some similarities with above-plume seawater communities. Similarities between plume and proximal habitat communities has also been observed by Olins et al. (2017), whereby intra-field (defined as within vent field between diffuse flows) and diffuse flow microbial communities were alike. In our study, similarities between plume and below-plume are likely derived by precipitation of mineral and microbial aggregates dragging plume microbes deeper below the plume as suggested by Dick et al. (2013). In





addition, internal wave induced turbulence causes vertical mixing along the slope of the Rainbow Ridge (van Haren et al., 2017), which may cause the plume and associated communities near the vent field to mix with ambient water communities leading to assemblage similarities. This indicates the plume and associated microbial processes could have a larger vertical footprint than previously observed, supporting suggestions by Olins et al., (2017) that proximal non-plume habitats have been overlooked. Interestingly, near-bottom water (and sediment) community assemblages were distinct from the below-plume and other water column communities. This could imply: 1) that there is little "fall out" from the plume at distance from the vent which is in agreement with sediment trap observations by Khripounoff et al. (2001), 2) plume specific bacteria die off due to lack of energy sources and DNA degrades before reaching the seafloor, 3) microbes are more abundant in the near-bottom waters, either naturally or through mechanical disturbance resuspending sediment during the coring process, outnumbering groups that have been mixed in from overlaying water. Despite the presence of a plume and precipitation, a barrier between the sea floor and the water column biotopes is present, consistent with global broad scale non-vent benthic-pelagic patterns (Zinger et al., 2011). According to Khripounoff et al. (2001) the fall-out from the Rainbow plume is spatially limited, as the extended chemical imprint on the sediment (reported by Cave et al. (2002), Chavagnac et al. (2005), and this study), is likely to have formed when the plume is in direct contact with the sediment during its vertical tidal migration. As the plume rises again, the associated distinct communities apparently resume dominance in the near-bottom water. Though Epsilonproteobacteria have been detected in Rainbow vent sediments comprising over 5 % of the sediment community (Lopez-Garcia et al., 2003), very few reads of this group in sediment samples were present in our study. Cave et al. (2002), observed chemical evolution of sediment composition with distance from source, thus we infer the dependence of sediment dwelling Epsilonproteobacteria on nearby plume precipitates, such as Cu, Zn and Cd. Additionally, DNA degradation rate can be 7 to 100 times higher in sediment than in the water column (Dell'Anno and Corinaldesi, 2004). Therefore, although our results suggest no microbial plume community imprint on the sediment, we cannot rule out short lived episodic community changes when the plume is in contact with the sediment.



## 4.3    Geochemical gradients within the hydrothermal plume

Analysis of SPM in water samples taken along the flow path of the plume, as well as off the flow path, showed conspicuous trends of elements, reflecting the chemical evolution of the plume as it drifts away from its hydrothermal source.

The chalcophile elements (Cu, Co and Zn) were found to have the highest element/Fe molar ratios closest to the vent site, indicating either rapid removal from the hydrothermal plume or removal from the solid phase as the plume drifts away from the vent site. Using SEM-EDS, it was demonstrated that at the proximal downstream stations mainly Fe-sulfides were found, whereas Fe-(oxyhydr)oxides were found further downstream. This suggests that chalcophile elements are mainly present in the form of sulfide mineral particles at the proximal stations, which are entrained in the flow of hydrothermal water emanating from the Rainbow vents and subsequently rapidly lost by settling from the plume in sulfide-bearing phases, while a large portion of Fe remains is suspension (Cave et al., 2002; Edmonds and German, 2004), consistent with decreasing concentrations of Cu, Zn and Cr in sediment recovered from the Rainbow area with increasing distance to the vent site (Cave et al., 2002).

The oxyanions (V and P) showed constant element/Fe molar ratios with increasing distance away from Rainbow, suggesting co-precipitation with Fe as oxyhydroxides (Edmonds and German, 2004). No additional uptake of these elements was observed with increasing distance from the vent field (German et al., 1991), since these elements are scavenged initially in significant amounts during the buoyant plume phase (Cave et al., 2002).

The trend shown by Mn/Fe molar ratios can be attributed to the slower oxidation kinetics of Mn (Cave et al., 2002). It takes longer for reduced Mn to be oxidised than it would for Fe, resulting in an increase in particulate Mn with increasing distance from the Rainbow hydrothermal vent field, which subsequently settles out from the plume as Mn-oxyhydroxides (Cave et al., 2002).

The observed positive relationship between the REEs and Fe is indicative of continuous scavenging of these elements from the ambient seawater onto Fe-oxyhydroxides (Caetano et al., 2013; Edmonds and



German, 2004). Therefore, the highest element/Fe molar ratios were observed away from the Rainbow
hydrothermal vent site, where Fe-(oxyhydr)oxides are dominant more distal to the vent site.
The Ca/Fe molar ratios vary between 0 and 14 for the stations downstream of the Rainbow hydrothermal
vent, but are higher at the distant downstream station 47 and 49 and upstream stations 28 and 40.
Especially at station 40, located on the Rainbow Ridge, the Ca/Fe molar ratio is significantly higher than
at the other stations. This is in line with observations by Khripounoff et al. (2001) and Cave et al. (2002)
who also found that the relative Ca concentration in settling particles and the sediments is lower close the
Rainbow vent field and increases as the Fe concentration decreases when the plume disperses. Since Ca
is naturally present in high abundances in pelagic skeletal carbonate which rains down from the overlying
water column and Fe is mainly present as a hydrothermal component the Ca/Fe molar ratio could be an
indicator for the extent of the hydrothermal influence. The high molar ratio at station 40 would then
suggest that this station is hardly or not at all influenced by the hydrothermal plume, whereas station 28,
47 and 49 are, as expected, influenced in more moderate degrees compared with the stations directly
downstream of Rainbow.

## 4.4 Microbial gradients within the hydrothermal plume

The microbial plume community composition and diversity altered with distance from the plume source
showcasing a horizontal heterogeneity within the plume. Despite dilution, the vent associated group
Epsilonproteobacteria (specifically the most common genus *Sulfurimonas*), appeared to dominate the
community composition. This is likely due to its flexibility to exploit many sulfur compounds as electron
donors, and oxygen and nitrate as acceptors (Nakagawa et al., 2005), making them suitable inhabitants of
dynamic environments (Huber et al., 2003). It is unclear from the relative abundance data obtained,
whether Epsilonproteobacteria dominate by rapid reproduction or if other groups decline in abundance.
However, it is evident that Epsilonproteobacteria remain competitive or outcompete other competitors
such as generalists Gammaproteobacteria that are often vent associated (i.e. SUP05). It is unlikely that
this pattern is caused by entrainment of Epsilonproteobacteria from background seawater over time. This
is based on the lack of significant presence of Epsilonproteobacteria in above-plume water and at remote



station 13, and reduced mixing that neutrally buoyant plumes generally experience (McCollom, 2000). This is further supported by the increasing uniqueness of the plume community with distance from the source, suggesting that mixing and entrainment between downstream biotopes is negligible.

The neutrally buoyant plume is likely too chemically enriched for non-adapted microbial taxa to thrive, and consequently are outcompeted by groups that can benefit from or tolerate the chemical nature of the plume. Therefore, it is likely that less specialised groups die out due to lack of appropriate resources and interspecies competition, as indicated by the decline in biodiversity with age of plume (distance) directly mirroring the increasing dominance of Epsilonproteobacteria, a group already known to influence community structures (Opatkiewicz et al., 2009). In addition, the decrease in concentration of particulate matter may influence microbial diversity (Huber et al., 2003). Temporal succession has been observed within plume environments by Sylvan et al., 2012 and Reed et al., 2015, driven by metabolic energy yield and concentration of the electron donors. We propose that the patterns in our study reflect ecological succession (Connell and Slaytor, 1977) within the plume with change in microbial communities resulting in a low diversity, climax plume community. At the distant stations 47 and 49, the community was less dominated by Epsilonproteobacteria and more diverse, indicating a gradual return to what is likely a non-plume influenced state of the microbial community. The wide range of correlations within and between microbial classes and water properties, i.e. ranging from chemical to physical variables (Fig. S4), indicates a complex array of community drivers within the plume.

In contrast to our results, Sheik et al. (2015) and Djurhuus et al. (2017), observed decreasing Epsilonproteobacteria abundance within hundreds of metres from the source in the rising, buoyant portion of plumes generated by Indian Ocean and South Pacific vents. Interestingly, in our results Epsilonproteobacteria were least dominant in the freshest neutrally buoyant fluid at the station closest to the Rainbow vent site. It is likely that entrainment of other microbial groups within the rising portion of the plume dilutes the contribution by this group. However, Huber et al., 2003 suggested that Epsilonproteobacteria, thrive in weaker diffuse flow due to lower temperature and great electron acceptor availability. A sampling design to follow the continuity of the plume from the buoyant to the neutrally buoyant portion would be a suitable approach to fully trace the evolution of the plume from the orifice to



full dilution. However, the term full dilution is ambiguous as it is unknown exactly how far the plume influences the water properties and how far the plume associated bacteria will follow, adding water column microbial community heterogeneity beyond our study spatial extent.

### 4.5    Possible effects of SMS mining plumes

Mining of SMS deposits will create additional plumes generated by activities of mining vehicles (resuspension) and by the discharge of solids from the surface vessel (discharge plume). It is yet unknown how these plumes will affect the ecosystem at active and inactive hydrothermal vent sites. Our study showed the influence of a natural hydrothermal plume on its environment up to 25 km away from its source and it was shown how a natural plume has a strong impact on the pelagic microbial and chemical composition, suggesting that mining plumes may cause similar changes to the background T0 state.

Excavation of SMS will cause removal of habitat by substrate extraction and resuspension of surface sediments. While large particles in the resulting plume are expected to stay close to the seafloor and eventually settle, smothering fauna in the immediate surroundings (Jones et al., 2018), smaller particles will disperse further, potentially invoking effects on a larger spatial scale. Another main concern is the discharge of mining waste, consisting of very fine unconsolidated particles, toxic metals and metal compounds (Weaver et al., 2018). Modelling the behaviour of the discharge plume generated by the proposed Solwara 1 SMS mining has shown that these plumes can extend up to 10 km from the mining site, resulting in a deposit thickness of up to 50 cm within 1 km of the discharge site (Gwyther et al., 2008; Boschen et al., 2013), smothering benthic fauna (Boschen et al., 2013; Weaver et al., 2018; Jones et al., 2018). Besides the impact caused by settling of particles from the excavation and discharge plumes, there is also the possible input of nutrients and toxins to otherwise nutrient- and toxin-poor systems, for example from oxidation of newly exposed sulfides and the subsequent release of heavy metals in the water column (Jones et al., 2018; Weaver et al., 2018).

The extent of the local impact of deep sea mining will depend on the location where the mining takes place. At an active site like the Rainbow hydrothermal vent field, we showed that even in the distant



plume (25 km away from Rainbow) hydrothermal plume microbiota dominate. When a mining discharge plume at an active hydrothermal vent field would be merged with the natural plume, the local effects might be minimal since microbial communities are already adapted to the metal-rich environments (Gwyther et al., 2008). However, a mining plume consisting of a dense suspension of bottom sediment and fine-grained metal sulfides is expected to support an altered microbial community in terms of abundance and composition, impacting the hydrothermal plume community. Moreover, the effects over larger spatial scales could be multiplied because of the increased export of electron donors by mining activities. Reed et al. (2015), who studied a hydrothermal plume in the Lau basin, have shown that the export of the chemolithoautotrophs from a plume increases with increasing availability of electron donors. Dispersion of chemolithoautotrophs is variable between groups depending on the energetics of their metabolisms, for example, methanotrophs which could disperse more than 50 km, are likely to disperse further than sulfur oxidisers (Reed et al., 2015). Increased export of microbial biomass from plumes may have impact on other marine systems which are hospitable to chemolithotrophs, such as oxygen minimum zones (Dick et al., 2013) and to higher trophic levels (Phillips, 2017). At inactive sites the effect on the background fauna is also potentially large since these are not adapted to the heavy metal rich environments and the discharge plume could prove to be toxic to the fauna (Boschen et al., 2013), possibly affecting organisms at all levels of the food chain (Weaver et al., 2018). In addition, in case of multiple plumes at different depths due to stratification and vertical migration due to tidal regimes, the impacts may not be confined to a single depth band and may affect a large part of the water column, including other habitats, such as benthic habitats.

## 5    Conclusion

Our results demonstrate geochemically enriched plumes provide a dynamic habitat that is conducive to ecological changes in a short time span. Combining microbial and chemical analysis has proven to be a sensitive tool which enabled us to trace the hydrothermal plume beyond 25 km downstream from the vent source and also upstream of the Rainbow vent site, implying that the influence of the hydrothermal vent on the surrounding environment may reach further than previously thought. The neutrally buoyant plume



was chemically enriched which spawned a distinct microbial biotope which was dominated by vent
associated species. As the plume aged and dispersed we observed alteration of the chemical composition
and microbial community composition of the plume, showcasing a horizontal heterogeneous plume.
Overall we have shown that a hydrothermal plume acts as a unique chemically enriched environment
where distinct and variable habitats are present.

## Author contribution

GD, HDS, and FM conceptualised the study and undertook data collection. SH and DP undertook sample
processing and analysis with contributions from and under the supervision of FM, GD, GJR, HDS, JvB
and HW. SH and DP wrote the manuscript with contributions from all co-authors.

## Competing interests

The authors declare that they have no conflict of interest.

## Acknowledgements

This study was carried out in the framework of the TREASURE (Towards Responsible ExtrAction of
SUbmarine REsources) project, funded (grant number 13273) by the Applied and Engineering Sciences
(AES) domain of the Netherlands Organisation for Scientific Research (NWO) and by partners from the
Dutch maritime industry. Topsector Water, a collaborative effort of Dutch industry, academia and
government, funded ship time. We thank Evaline van Weerlee for assistance in DNA extraction and
Patrick Laan for assistance in the chemical analysis of the collected samples. We also thank the crew and
captain of the RV *Pelagia*, as well as NIOZ technicians for their essential assistance during cruise
64PE398. SH received funding from the Blue Nodules project, EC grant agreement. 688785. DP is
supported by the Natural Environmental Research Council [grant number NE/N012070/1]. HdS received



funding from TREASURE. FM is supported financially by the Innovational Research Incentives Scheme
of the Netherlands Organisation for Scientific Research (NWO-VIDI grant 016.161.360).

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



## Figures and tables

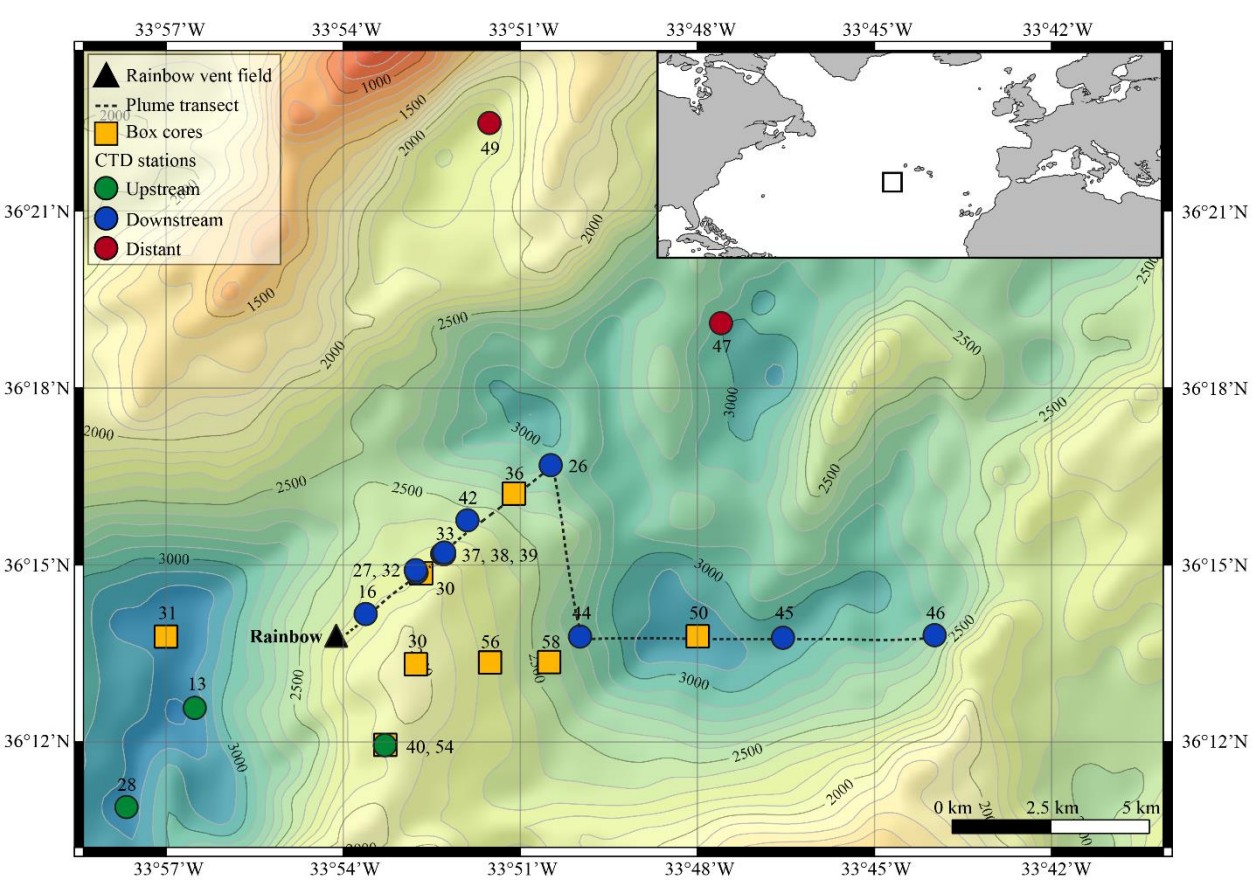

*Figure 1: Mid Atlantic Ridge bathymetry (EMOD) with Geographical location (inset), showing sampling methods and locations depicted.*




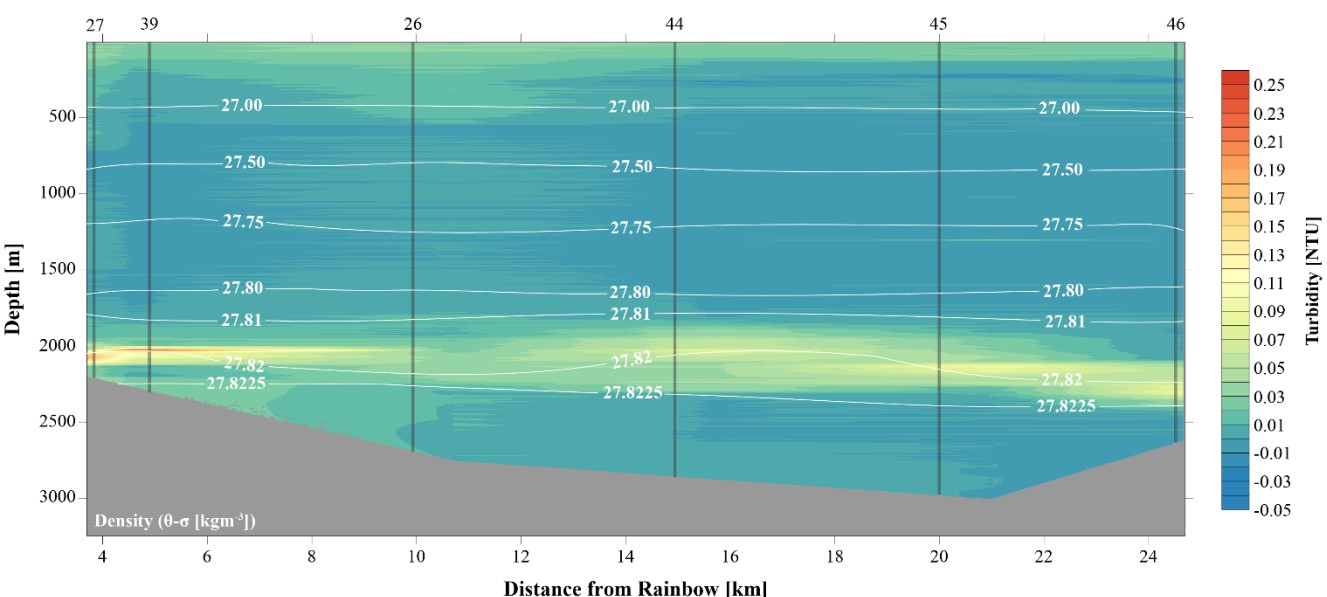


*Figure 2: Transect along main plume path showing turbidity in the water column. The plume is indicated by highest turbidity values and disperses away from the Rainbow vent field.*


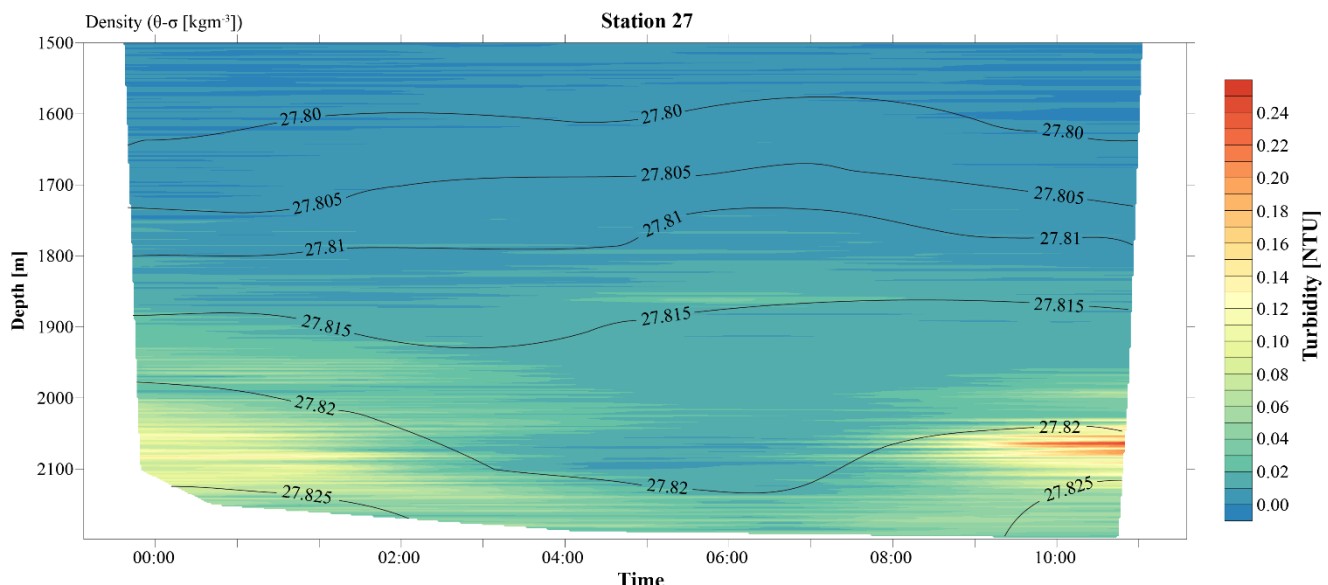


*Figure 3: 12 hour CTD YOYO casts at station 27 showing the temporal evolution of the hydrothermal plume over a tidal cycle.*





Figure 4: *(a) NMDS ordination showing all water samples based on their resemblance in chemical composition.*
*(b) NMDS ordination showing all plume samples from the downstream stations based on their resemblance in*
*chemical composition.*



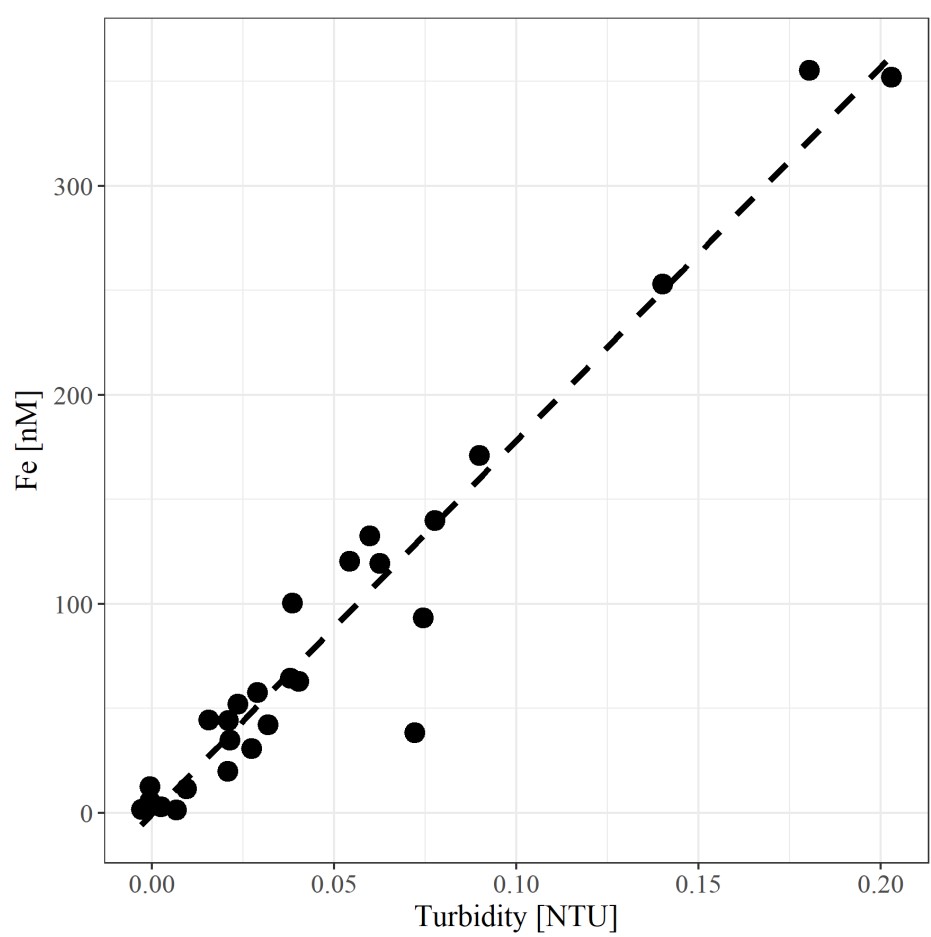


*Figure 5: Relationship between measured turbidity and molar concentration of iron.*







*Figure 6: Relationship between copper (a), vanadium (b), yttrium (c) and tin (d) to iron.*






*Figure 7: Element to iron molar ratios. Plume samples of upstream, downstream and distant stations. Downstream*
*stations follow the main path of the plume. Fig. 7a) shows the element/Fe molar ratios of the chalcophiles (Co, Cu*
*and Zn), b) shows the ratios of Mn and the oxyanions (P and V), c) displays the ratios of REE, d) the ratios of Al,*
*In, Ni, Pb, Sn, Ti and U and e) shows the Ca/Fe molar ratio.*

802





*Figure 8: Non-metric multidimensional scaling plot of the microbial community composition of all samples based on Operational Taxonomic units. Similarity groupings are based on group average clustering. "No plume" is representative of samples collected from station 13, where there was no indication of a plume.*





*Figure 9: Non-metric multidimensional scaling plot of the microbial community composition of all water column samples based on Operational Taxonomic units. Plume and below plume depths from Station 13 were excluded.*





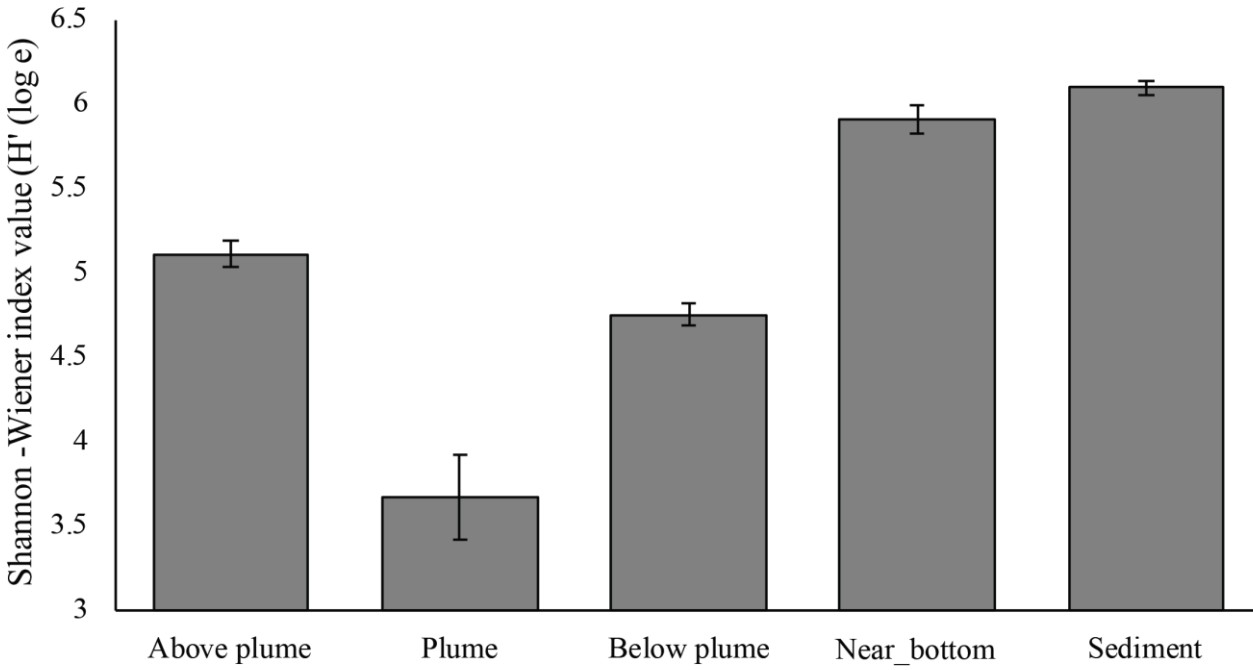

812

*Figure 10: Mean Shannon-Wiener diversity index for microorganisms in each biotope. Error bars represent ±SE*

814





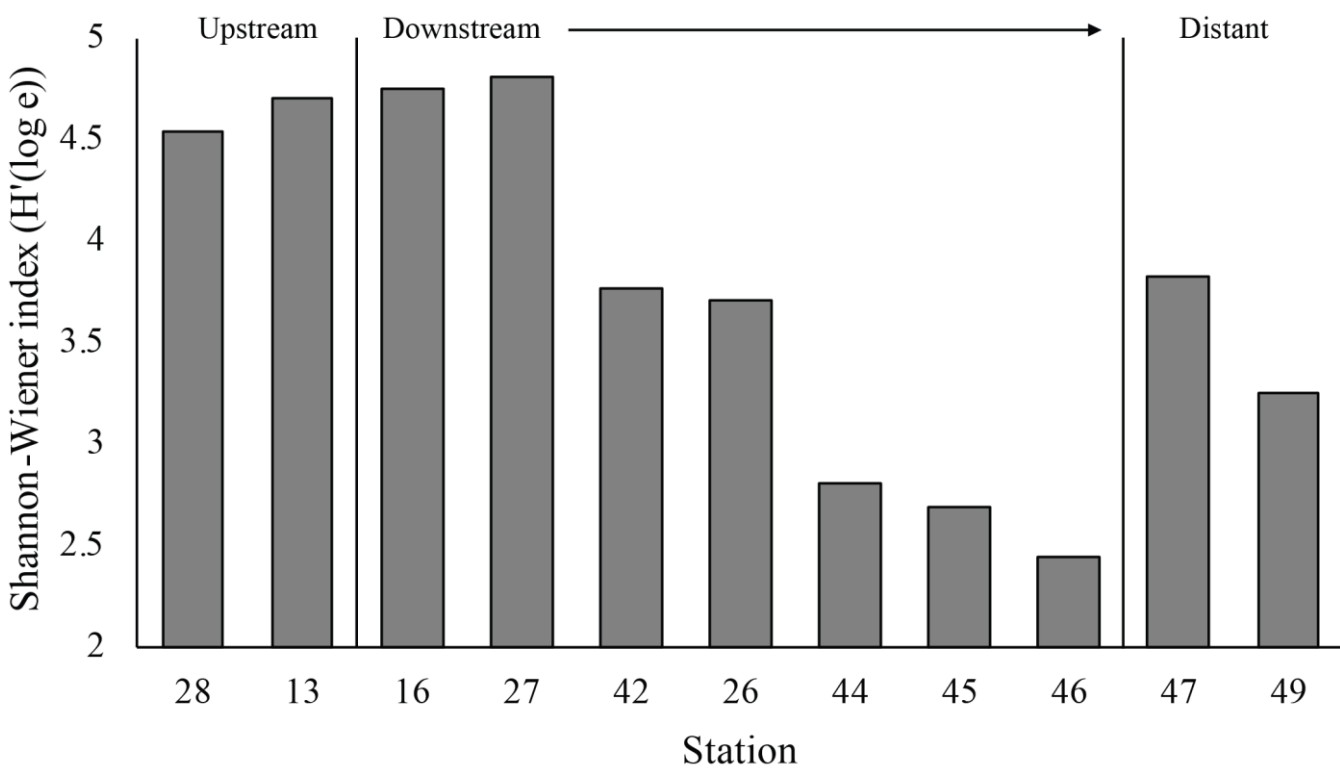

Figure 11: Shannon-Wiener index values for microorganisms in each plume sample taken.





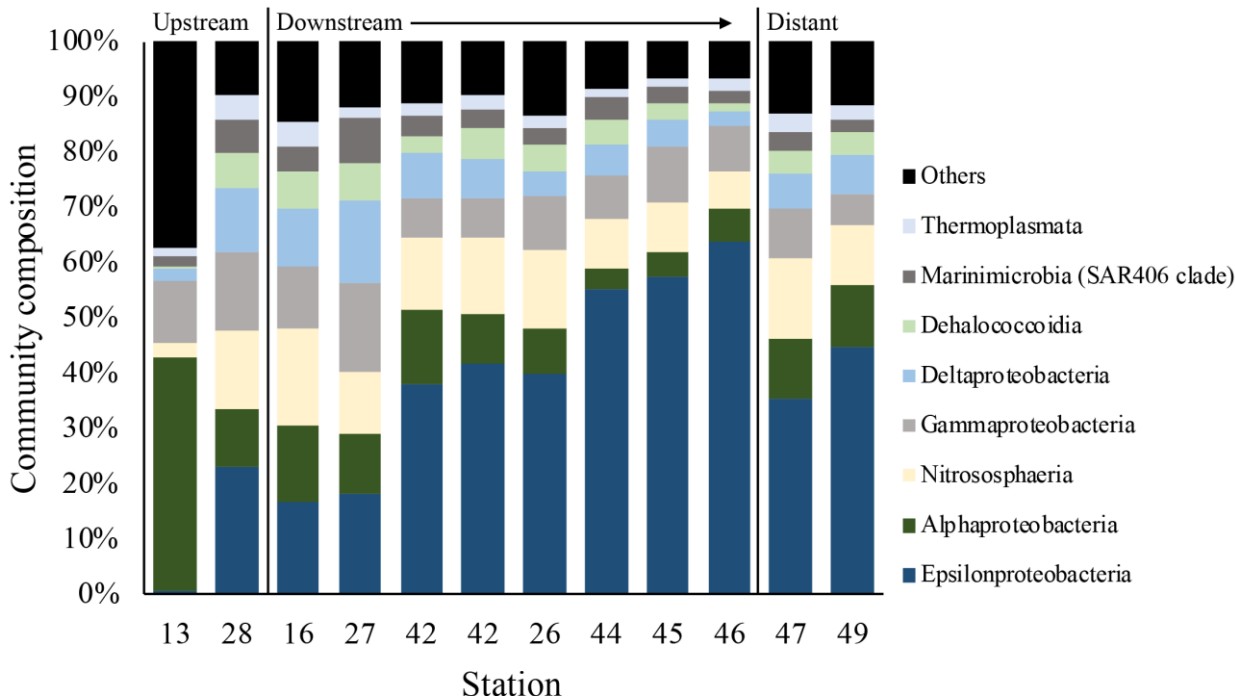

*Figure 12: Microbial community composition in the plume samples as a percentage of the dominant class groups in accordance with the SIMPER results.*





*Table 1: Meta-data of samples taken.*

| Station | Biotope | Sample type | Depth(m) | C/N | Microbiology | SPM | (Trace) metals |
|---|---|---|---|---|---|---|---|
| 30 | Sediment and near-bottom water | Box core | 1970 | | x | | |
| 31 | Sediment and near-bottom water | Box core | 3190 | | x | | |
| 33 | Sediment and near-bottom water | Box core | 2223 | | x | | |
| 36 | Sediment and near-bottom water | Box core | 2857 | | x | | |
| 50 | Sediment and near-bottom water | Box core | 3157 | | x | | |
| 54 | Sediment and near-bottom water | Box core | 2129 | | x | | |
| 56 | Sediment and near-bottom water | Box core | 2198 | | x | | |
| 58 | Sediment and near-bottom water | Box core | 2514 | | x | | |
| 13 | Above plume | CTD | 125 | x | x | | |
| 13 | Below plume | CTD | 3220 | x | x | | |
| 13 | Plume | CTD | 2000 | x | x | | |
| 16 | Plume | CTD | 1944 | | x | | |
| 16 | Above plume | CTD | 998 | | x | | |
| 26 | Below plume | CTD | 2756 | x | x | x | x |
| 26a | Plume | CTD | 2150 | x | x | x | x |
| 26b | Plume | CTD | 2000 | | | x | x |
| 26 | Above plume | CTD | 999 | | x | x | x |
| 27 | Below plume | CTD | 2191 | | x | | x |
| 27a | Plume | CTD | 2077 | | x | | x |
| 27b | Plume | CTD | 1996 | | | | x |
| 27 | Above plume | CTD | 994 | | x | | x |
| 28 | Below plume | CTD | 3170 | x | x | x | x |
| 28 | Plume | CTD | 1975 | x | x | x | x |
| 32a | Plume | CTD | 2192 | | | x | |
| 32b | Plume | CTD | 2088 | | | x | |
| 37 | Plume | CTD | 2190 | | | | x |
| 38 | Plume | CTD | 2040 | | | | x |
| 39 | Plume | CTD | 2019 | | | | x |
| 40 | No plume | CTD | 2120 | | | | x |
| 42a | Plume | CTD | 2291 | x | x | x | x |
| 42b | Plume | CTD | 2209 | x | x | x | x |
| 42c | Plume | CTD | 2037 | | | x | x |
| 42 | Above plume | CTD | 999 | x | x | x | x |
| 44 | Below plume | CTD | 2623 | x | x | | |
| 44a | Plume | CTD | 2202 | | | x | x |
| 44b | Plume | CTD | 2002 | x | x | x | x |
| 44 | Above plume | CTD | 995 | | x | | |
| 45 | Below plume | CTD | 3004 | x | x | | |
| 45a | Plume | CTD | 2166 | | | x | x |
| 45b | Plume | CTD | 2002 | x | x | x | x |
| 45 | Above plume | CTD | 996 | | x | | |
| 46 | Below plume | CTD | 2622 | x | x | | |
| 46a | Plume | CTD | 2280 | x | x | x | x |
| 46b | Plume | CTD | 2145 | | | x | x |
| 46 | Above plume | CTD | 1000 | | x | | |
| 47 | Below plume | CTD | 2850 | x | | | |
| 47 | Plume | CTD | 2200 | x | x | | x |
| 49a | Plume | CTD | 2260 | x | x | x | x |
| 49b | Plume | CTD | 1902 | | | x | x |





*Table 2: Primers used for sequencing.*

| | Forward | | Reverse | | | |
|---|---|---|---|---|---|---|
| Primer name | Primer sequence 5'-3' | Primer name | Primer sequence 5'-3' | Ratio in mix | Reference |
| Arch-0519-a-S-1 (universal) | CAGCMGCCGCGGTAA | Bact-0785-b-A-18 (universal) | TACNVGGGTATCTAATCC | 3/9 + 3/9 | Klindworth et al. 2012 |
| Bact-0519F (targets WS6, TM7, OP11 | CAGCAGCATCGGTVA | | | 1/9 | This paper |
| Nano-0519F (targets Nanoarchaea) | CAGTCGCCRCGGGAA | Nano-0785R (targets Nanoarchaea) | TACNVGGGTMTCTAATYY | 1/9+1/9 | This paper |







*Table 3: SIMPER similarity results of each biotope at class level. ** undefined class.*

| Biotope | Average similarity (%) | Class | Average proportion (%) | Average similarity | Sim/SD | Contribution (%) | Cumulative % |
|---|---|---|---|---|---|---|---|
| Above plume | 82.34 | Nitrososphaeria | 27.10 | 22.79 | 4.61 | 27.67 | 27.67 |
| | | Alphaproteobacteria | 18.34 | 15.22 | 4.15 | 18.49 | 46.16 |
| | | Gammaproteobacteria | 13.44 | 11.58 | 5.52 | 14.07 | 60.23 |
| | | Deltaproteobacteria | 10.67 | 8.46 | 3.38 | 10.27 | 70.50 |
| | | Marinimicrobia (SAR406 clade) | 8.22 | 6.96 | 6.07 | 8.46 | 78.96 |
| | | Dehalococcoidia | 6.38 | 5.69 | 9.19 | 6.91 | 85.87 |
| | | Thermoplasmata | 2.63 | 2.26 | 5.68 | 2.74 | 88.61 |
| | | Acidimicrobiia | 2.13 | 1.89 | 8.62 | 2.30 | 90.91 |
| Plume | 76.74 | Epsilonproteobacteria | 39.59 | 30.29 | 2.53 | 39.47 | 39.47 |
| | | Nitrososphaeria | 12.16 | 10.32 | 4.05 | 13.45 | 52.92 |
| | | Gammaproteobacteria | 9.69 | 7.92 | 4.71 | 10.32 | 63.23 |
| | | Alphaproteobacteria | 9.23 | 7.22 | 2.44 | 9.40 | 72.64 |
| | | Deltaproteobacteria | 7.60 | 5.56 | 2.75 | 7.25 | 79.88 |
| | | Dehalococcoidia | 4.57 | 3.55 | 2.58 | 4.63 | 84.51 |
| | | Marinimicrobia (SAR406 clade) | 4.02 | 3.07 | 3.83 | 4.00 | 88.51 |
| | | Thermoplasmata | 2.56 | 1.94 | 3.39 | 2.53 | 91.04 |
| Below plume | 77.94 | Nitrososphaeria | 22.35 | 16.60 | 3.29 | 21.30 | 21.30 |
| | | Alphaproteobacteria | 13.26 | 11.43 | 5.18 | 14.67 | 35.97 |
| | | Deltaproteobacteria | 10.88 | 9.25 | 8.31 | 11.87 | 47.84 |
| | | Gammaproteobacteria | 10.60 | 8.89 | 7.78 | 11.40 | 59.24 |
| | | Epsilonproteobacteria | 9.65 | 7.18 | 2.50 | 9.22 | 68.46 |
| | | Dehalococcoidia | 7.84 | 6.97 | 7.89 | 8.95 | 77.40 |
| | | Marinimicrobia (SAR406 | 6.32 | 4.49 | 2.31 | 5.76 | 83.16 |
| | | Thermoplasmata | 4.69 | 3.04 | 2.20 | 3.90 | 87.07 |
| | | Phycisphaerae | 1.97 | 1.75 | 7.60 | 2.24 | 89.31 |
| | | Planctomycetacia | 2.03 | 1.50 | 2.96 | 1.93 | 91.23 |
| Near-bottom water | 75.71 | Gammaproteobacteria | 20.79 | 16.77 | 3.18 | 22.15 | 22.15 |
| | | Nitrososphaeria | 16.90 | 13.54 | 3.79 | 17.89 | 40.04 |
| | | Alphaproteobacteria | 15.55 | 13.25 | 5.47 | 17.50 | 57.54 |
| | | Deltaproteobacteria | 6.68 | 5.89 | 5.99 | 7.78 | 65.32 |
| | | Oxyphotobacteria | 5.93 | 4.04 | 2.18 | 5.34 | 70.66 |
| | | Dehalococcoidia | 4.08 | 2.99 | 2.50 | 3.95 | 74.61 |
| | | Phycisphaerae | 3.72 | 2.57 | 2.03 | 3.40 | 78.01 |
| | | Thermoplasmata | 2.47 | 1.70 | 2.25 | 2.24 | 80.25 |
| | | Acidimicrobiia | 2.06 | 1.61 | 2.72 | 2.13 | 82.38 |
| | | Bacteroidia | 2.15 | 1.57 | 1.85 | 2.07 | 84.45 |
| | | Marinimicrobia (SAR406 clade) | 1.75 | 1.24 | 2.17 | 1.64 | 86.09 |
| | | OM190 | 1.64 | 1.14 | 2.02 | 1.51 | 87.60 |
| | | Planctomycetacia | 1.40 | 1.09 | 2.76 | 1.44 | 89.04 |
| | | Epsilonproteobacteria | 1.71 | 0.85 | 1.08 | 1.12 | 90.16 |
| Sediment | 82.51 | Gammaproteobacteria | 29.67 | 27.17 | 8.51 | 32.93 | 32.93 |
| | | Alphaproteobacteria | 13.98 | 12.44 | 4.88 | 15.07 | 48.01 |
| | | Deltaproteobacteria | 11.98 | 10.98 | 10.24 | 13.30 | 61.31 |
| | | Nitrososphaeria | 7.73 | 5.69 | 3.74 | 6.90 | 68.21 |
| | | Phycisphaerae | 5.46 | 5.01 | 7.85 | 6.07 | 74.28 |
| | | Dehalococcoidia | 3.35 | 2.48 | 2.58 | 3.01 | 77.29 |
| | | BD2-11 terrestrial group | 2.36 | 1.91 | 2.90 | 2.31 | 79.60 |
| | | Subgroup 22 (Acidobacteria) | 2.10 | 1.74 | 3.22 | 2.11 | 81.71 |
| | | OM190 | 2.09 | 1.50 | 5.50 | 1.81 | 83.53 |
| | | Nitrospira | 1.79 | 1.49 | 3.68 | 1.80 | 85.33 |
| | | Bacteroidia | 1.91 | 1.48 | 3.66 | 1.79 | 87.12 |
| | | Acidimicrobiia | 1.58 | 1.24 | 2.84 | 1.50 | 88.62 |
| | | Thermoanaerobaculia | 1.41 | 1.07 | 3.25 | 1.30 | 89.92 |
| | | Gemmatimonadetes** | 1.57 | 1.06 | 1.56 | 1.28 | 91.21 |
