# Peer review of "Patterns of (trace) metals and microorganisms in the Rainbow hydrothermal vent plume at the Mid-Atlantic Ridge"

_Biogeosciences, 2019_

## Referee Comment (RC1) · Valérie Chavagnac (Referee) · 25 Jul 2019

I was quite enthusiastic to accept this review as I have been working at the Rainbow vent site. Considering the interest of mining companies to exploit the deep-sea hydrothermal deposit, i thought that the submitted paper will provide key information on the close interaction between microbial diversity and the environmental conditions. The topic of this paper can be very useful for the scientific community.

Unfortunately, I had a hard time to review the paper as it lacks general informations, some references are missing and the geochemical data are missing. Therefore, I cannot provide an appropiate review of this paper considering that half of the paper is deal-

ing with the environmental changes along the missing gradient between hydrothermal fluid an seawater. I am a geochemist with limited expertise with microbiology. So I leave this section to other review. I stopped the review at mid-way as I cannot see their data and how they have been acquired (the methodology is poorly described). I cannot provide an objectif report.

As such, I cannot accept this paper. The topic is still valuable to the scientific community but the authors need to improve significantly their manuscrit.

Most of my comments are directly indicated on the manuscript pdf.

Sincerely yours, Valérie Chavagnac

Please also note the supplement to this comment:
https://www.biogeosciences-discuss.net/bg-2019-189/bg-2019-189-RC1-supplement.pdf

—————————————————————

---

## Referee Comment (RC2) · Anonymous Referee #2 · 30 Jul 2019

Haalboom and collaborators conducted a multidisciplinary (geochemical/microbiological) study at the Rainbow vent site. They collected samples from different sites around Rainbow, at downstream stations, distal downstream stations, and upstream stations, and included different depths (above the plume, inside the plume and below the plume), as well as near-bottom water and sediment. The geochemical characteristics in the particulate phase were studied. The study was focused on trace metals and Rare Earth Elements. In parallel, they studied microbial communities and their variations in the different biotopes. These results are potentially very interesting and could provide new information for the knowledge and understanding of hydrothermal sites and their geochemical and microbiological characteristics.

[Figure]

However, the link between the different geochemical parameters is not sufficiently detailed. What does the combination of Rare Earth Elements and trace metals really bring to the story? Similarly, the link between geochemical parameters and microbial communities is not sufficiently exploited. For example, one of the major results that should have been discussed is Figure S4, which shows the correlations between environmental variables and classes of microorganism. It is only indicated that there is "a complex array of community drivers within the plume". Moreover, the authors claim that their study represents a T0 before mining activities, but I am not convinced by the analogy between the 2 types of plumes. Indeed, if the geochemical characteristics could be similar, the temperature, density, and microbial communities will be totally different.

Specific comments:

Title: I am not convinced that the results show the successional patterns of trace metals and microorganisms and I would recommend to remove the word "Successional".

Sampling (p 6): The sampling strategy seems confusing to me. Why several stations were sampled at the same location? What is the difference between these stations? The differences observed for the same parameter among the stations are not discussed. SPM, trace metals, and the microbial community are not systematically sampled at the same location. For example, stations 37, 38, and 39 were only sampled for trace metals. Is there any explanation why the different depths of each station were not systematically sampled for all parameters? It is indicated that intermittent water samples were taken for nutrients, but no information is reported on Table 1. For suspended particulate organic matter, I assume the authors refer to C/N on Table 1. No information is given for the analyses of nutrients and POC/PON. I understand that coring sites were constrained by the coring substrate, but why was not CTD deployed at each coring site?

SPM analyses (p 7): I would have liked to see the values of blank filters and the associated uncertainties as well as the average percentage they represent. Please write what SEM and EDS mean.

Chemical analysis (p 6): This section is missing some important information and is much less detailed that the following one. Were the filters acid-cleaned before use? What are the values for the filter blanks? Were procedural blanks performed? Which certified reference material was used to asses the accuracy of the analyses?

Statistics (p 9): For the biodiversity index, the authors should be consistent along the ms. With the name of the index (Shannon-Wiener vs. Shannon).

Water column characteristics (p 10): Using the T-S diagram, the authors identified 3 water masses. However, the hydrography of the area is certainly more complex than that, as shown in the article by Jenkins et al (2015, http://dx.doi.org/10.1016/j.dsr2.2014.11.018), even if this later study was located further south.

Enrichment of trace metals compared to the ambient seawater (p 11). In addition to the enrichment factors, I would have liked to see vertical profiles of the absolute values of trace metals and the range of variations. How was the "clear water" defined?

Geochemical gradients (p 12): Fe was found to be linearly correlated to the turbidity with a $R^2$ higher than 93%. What was the p value? In the text, it is written that the chalcophile elements Co, Cu, and Zn are shown on Fig. 6A, but only Cu is shown. Same for V and P for Fig. 6B and REEs for Fig. 6C, where only V and Y are shown. Similarly, in the text, Mn, Al, Ni, In, Pb, Ti, and U are referred to Fig. 6D while Sn is shown on this figure. Line 301: the authors state that Zn/Fe ratio is elevated at stations 37, 39, and 44. This is also the case at station 40, and is not discussed in the text. Line 302: on Fig. 6B, the relation between V and Fe indeed looks linear, but the axes are drawn with a logarithmic scale, which means that the relationship is not linear but polynomial. The V:Fe ration is not more or less constant and display values from 0.005 to ∼ 0.012 (please change also on line 462). It is the same for the REEs.

Microbial assemblages (p 13): Line 316: please replace "above plume" by "no plume" Line 317: please replace "which clustered distinctly from each other and from plume and below-plume communities" by "which clustered distinctly from each other and from plume, below-plume, and above-plume communities" Line 318: please replace "sediment and near-bottom water samples have communities that are very dissimilar from the overlying water column samples" and "sediment, near-bottom water, and no-plume samples have communities that are very dissimilar from the overlying water column samples"

Univariate biodiversity (p 13): Data used for Fig. 10 and Fig. 11 is slightly confusing. In Fig. 10, the value for diversity index in the plume is about 3.5 with SE lower than 0.5. In fig. 11, the values for samples in each plume vary from less than 2.5 to higher than 4.5. So I am wondering if the value in Fig. 10 corresponds to the average value of the data in Fig. 11 or not.

Plume influence on the water column chemical and microbial make-up (p 16-17): A table with the range of variation of the literature values would be useful. Line 408: please specify here what you mean with oceanic water masses. Line 411: please specify what you mean with SUP05 Line 442-443: the authors infer the dependence of sediment dwelling Epsilonproteobacteria on nearby plume precipitates, such as Cu, Zn and Cd, but why only these 3 elements? This should be justified.

Geochemical gradients within the hydrothermal plume (p 19): The high Ca:Fe ratio at station 40 is explained by the non-influence of hydrothermal plume. Please add a reference for this statement.

Microbial gradients within the hydrothermal plume (p 20): The authors state that the dominance of Epsilonproteobacteria is likely driven by the strong chemical enrichment of the plume but when looking at Fig. S4, Epsilonproteobacteria is not within the group that is the most strongly positively correlated with trace metals. As I wrote above, this point would be very interesting to discuss as well as the other correlations. Lines

511-513: this statement is too speculative.

Figures and Tables: Fig. 1: Station 30 is indicated twice. Fig. 2: The X axis represents the distance from Rainbow. On Fig. 1, it looks like station 44 is located closer to Rainbow than station 26. Table 1: Could you indicate long-lat for each station?

---

## Author Comment (AC1) · 26 Nov 2019

Comments reviewer 1 (Valérie Chavagnac) BG-2019-189

We would like to thank Valérie Chavagnac for her efforts and input provided. We carefully went through all the comments and suggestions and have adjusted the manuscript according to the comments made. Below we provide descriptions of the adjustments made, addressing the reviewer's remarks.

The responses on the comments are given below, but please note the added supplement where the responses are given with proper formatting.

Note) Line numbers: First original manuscript, second revised manuscript

General comments:

1) "I thought that the submitted paper will provide key information on the close interaction between microbial diversity and the environmental conditions"

The aim of this study was to characterise the state of a hydrothermal plume before it is impacted by deep-sea mining to serve as a baseline study which will aid in monitoring of the impacts of deep-sea mining, as the situation after mining can then be compared to a state before mining. The plume is characterized in terms of geochemistry and the microbial assemblages as it disperses away from its source. It is not in the scope of this study to exploit the close interaction between the microbial diversity and the environmental conditions. We do agree we should have made this clearer at the start of the manuscript and have made adjustments in both the abstract and the introduction.

L21-24 (L21-L24): "Understanding how hydrothermal plumes can be characterised by means of geochemistry and microbiology as they spread away from their source and how they affect their surrounding environment may help in characterising the behaviour of the dilute distal part of chemically enriched mining plumes."

L36 (L41-43): Added: "This study of a hydrothermal plume serves as a baseline study to characterize the natural plume before the interference of deep-sea mining".

L103 (L105-109): "Whilst mechanic understanding of microbial and geochemical interactions in the plume would have required a different experimental setup, which was beyond the scope of the TREASURE project, this paper aims to contribute to knowledge of geochemical and biological heterogeneity in the surroundings of an SMS site, induced by the presence of an active hydrothermal plume, which should be taken into account in environmental impact assessments of SMS mining."

2) "It lacks general information, some references are missing and the geochemical data are missing" Based on the comments given in the rest of the manuscript general

information and missing references are added. Please see the comments below for more details.

A table with the full geochemical dataset (concentrations in pM, with precision in %) will be made public in PANGAEA when the manuscript is published and is also already available in the NIOZ data portal (https://dataverse.nioz.nl/dataverse/doi under DOI 10.25850/nioz/7b.b.s). We have added a table in the supplement (Table S2) showing part of the (trace) metal and REE data as we compare it to other work.

3) "I cannot see their data and how they have been acquired (the methodology is poorly described)" We have extended the methodology to better describe how the data have been acquired. The changes are shown at general comments 7 and 8 in more detail.

A table with the full geochemical dataset (concentrations in pM, with precision in %) will be made public in PANGAEA when the manuscript is published and is also already available in the NIOZ data portal (https://dataverse.nioz.nl/dataverse/doi under DOI 10.25850/nioz/7b.b.s). We have added a table in the supplement (Table S2) showing part of the (trace) metal and REE data as we compare it to other work.

4) Abstract: I find the abstract too vague and not enough information on what the authors have done during the course of their study. I suggest to reduce the first paragraph and to concentrate the text on the results and conclusions.

We did not reduce the first paragraph as we think it is important information as this study was done within the TREASURE project, which is related to deep-sea mining. However, we have made changes, focusing more on the results and conclusions.

L21-24 (L21-24): Changed "Understanding how natural hydrothermal plumes evolve as they spread away from their source and how they affect their surrounding environment may provide some analogies for the behaviour of the dilute distal part of chemically enriched mining plumes." to "Understanding how hydrothermal plumes can be characterised by means of geochemistry and microbiology as they spread away from their

source and how they affect their surrounding environment may help in characterising the behaviour of the dilute distal part of chemically enriched mining plumes."

L31-32 (L31-37): Expanded "...the neutrally buoyant plume stood out by its enrichments in (trace) metals and REEs, of which the concentrations changed as the plume aged", to "...the neutrally buoyant plume stood out by its enrichments in (trace) metals and REEs as e.g. Fe, Cu, V, Mn and REE were enriched by factors of up to ∼80, ∼90, ∼52, ∼2.5 and ∼40 respectively, compared to clear water samples taken at 1000 m water depth. The concentrations of these elements changed as the plume dispersed shown by the decrease of element/Fe molar ratios of chalcophile elements (Cu, Co, Zn), indicative of rapid removal from the hydrothermal plume or removal from the solid phase. Conversely, increasing REE/Fe molar ratios imply uptake of these elements from the ambient seawater onto Fe-oxyhydroxides."

5) Introduction: As it stands, by the end of the introduction, I don't have any clues on the methods that you will be using and for what. Please provide some additional information. We have provided additional information on the methods used.

L97-100 (L101-105): Changed "Geochemical and biological changes were tracked vertically in the water column and horizontally along the neutrally buoyant plume to study the heterogeneity in the background pelagic system that was influenced by the hydrothermal plume." to "Geochemical and biological changes were explored vertically in the water column and horizontally along the neutrally buoyant plume using HR-ICP mass spectrometry to determine the (trace) metal and REE content of the SPM and next generation sequencing methods were used to quantify the heterogeneity in the background pelagic system that was influenced by the hydrothermal plume."

6) Material and methods, study site: Some information are missing and are provided in German et al., 1996; Marques et al., 2006

In our opinion not much was mentioned in these papers what we did not mention yet in our setting description. We have added German et al. (1996) and Marques et al.

(2006) as additional references (L111 (L116); L114 (L120)).

7) Material and methods, suspended particulate matter analysis: Unclear on the procedure you applied. What has been done onboard and onshore. Please clarify.

It was mentioned what was done onboard (L156 (L165) "The subsamples were filtered on board over pre-weighed 0.4 $\mu$m polycarbonate filters." To better emphasize what we did on shore we changed L158 (L167) to "In the laboratory, the filters were freeze dried..."

L163-164 (L173-174): Added under which conditions the SEM was operated: "The SEM was operated under an acceleration voltage of 15 kV and a filament current of 1850 mA."

8) Material and methods, chemical analysis: Unclear what has been down onboard and on shore. Please provide additional information about the calibration of the instrument, the blank, the drift correction etc. Where is the table of results?

In order to make it more clear what was done onboard and onshore the following changes have been made: L166-167 (L178): "...water samples were filtered on board..." L170 (L181): "Filters were dried in the laboratory..."

L176 (L188-191): Added information of the procedural blanks in the geochemical analysis: "Furthermore, ten procedural blanks were performed. Half of them were empty acid-cleaned Teflon vials, the other five contained an acid-cleaned blank filter in order to correct for the dissolved filters. The blanks were subjected to the same total digestion method as described above."

L178 (L193-195): Added information about the calibration of the instrument: "The concentrations were calculated using external calibration lines made from a multi stick solution, which was prepared by mixing Fluka TraceCert standards for ICP. Rh was used as an internal standard for all elements."

L178 (L195-196): Added information about the drift measurements: "The machine drift

was measured before, half-way and after each series of samples and was monitored by using an external drift solution."

L178 (L196-200): Added information about the precision: "Precision (relative standard deviation (RSD)) of these analyses was generally <2 % for major- and trace metals, apart from 115In where the RSD values generally are between 4 % and 8 %, with maximum values going up to 12.48 %. For REE, the RSD values were generally <3 %, apart from a few measurements with RSD values reached maximums up to 12.48 %."

L178 (L200-201): Added information about the accuracy: "The accuracy could not be determined as no certified reference material was analysed."

L178 (L201-204): Added information on what the blanks were used for and how the true concentration was calculated: "The data of the samples was corrected for the dissolved filters by subtracting the average result of the five blank filters. Subsequently the data was recalculated to account for the dilution of the samples during the total digestion and the amount of seawater that was filtered to yield the true concentration of each element."

A table with the full geochemical dataset (concentrations in pM, with precision in %) will be made public in PANGAEA when the manuscript is published and is also already available in the NIOZ data portal (https://dataverse.nioz.nl/dataverse/doi under DOI 10.25850/nioz/7b.b.s). We have added a table in the supplement (Table S2) showing part of the (trace) metal and REE data as we compare it to other work.

Specific comments:

1) Abstract, P2, L30: "Both vertically in the water column and horizontally along the neutrally buoyant plume, geochemical and biological changes were evident as the neutrally buoyant plume stood out by its enrichments in (trace) metals and REEs, of which the concentrations changed as the plume aged." I find this sentence too vague to provide additional information compared to the literature. It would be much appreciated to

add some quantification on trace element concentration for example.

L31-32 (L31-37): Expanded ". . .the neutrally buoyant plume stood out by its enrichments in (trace) metals and REEs, of which the concentrations changed as the plume aged", to ". . .the neutrally buoyant plume stood out by its enrichments in (trace) metals and REEs as e.g. Fe, Cu, V, Mn and REE were enriched by factors of up to ∼80, ∼90, ∼52, ∼2.5 and ∼40 respectively, compared to clear water samples taken at 1000 m water depth. The concentrations of these elements changed as the plume dispersed shown by the decrease of element/Fe molar ratios of chalcophile elements (Cu, Co, Zn), indicative of rapid removal from the hydrothermal plume or removal from the solid phase. Conversely, increasing REE/Fe molar ratios imply uptake of these elements from the ambient seawater onto Fe-oxyhydroxides."

2) Abstract, P2, L34: ". . .the biodiversity appeared to reduce with distance away from the Rainbow hydrothermal vent field" What is this biodiversity change?

The change in biodiversity of the microbial background pelagic system was that it reduced with distance from the Rainbow hydrothermal vent field. Biodiversity was quantified into a univariate indice to quantify this reduction in diversity. L34 (L39): changed to ". . .univariate microbial biodiversity declined with distance away from the Rainbow hydrothermal vent field."

3) Abstract, P2, L36: What would be the connection with the impact of deep-sea mining?

L36 (L41-L43): Added: "This study of a hydrothermal plume provides a baseline study to characterize the natural plume before the interference of deep-sea mining".

4) Introduction, P2, L42: Add reference

L42 (L49-50): Added Cave et al. (2002) and Chavagnac et al. (2005) as references.

5) Introduction, P2, L44: Remove possible

L44 (L51): Removed possible.

6) Introduction, P3, L58: "Remove south of the Azores", change to "36°14" N on the MAR"

L58 (L65): Changed "south of the Azores" to "36°14" N on the MAR"

7) Introduction, P3, L59: "...it ejects one of the most prominent and persistent natural plumes on the MAR" Hydrothermal fluids at Rainbow are extremely enriched in Fer compared to other vent fields along the MAR. However, the substratum is not solely composed of basalt as it is elsewhere such as Menez Gwen, Lucky Strike etc. It would be valuable to provide additional information with some references.

In the following paragraph of the introduction we mention that it is shown that the host rock influences the hydrothermal fluid composition (see L69-70 (L78-79): "..., that the underlying host rock influences the hydrothermal fluid composition...".) Furthermore, it is mentioned in the setting description that the basement rocks are different compared to most other sites, L113-122 (L118-125): "The vent field, which is approximately 100 by 250 m in size, is underlain by a basement composed of ultramafic rocks (Edmonds and German, 2004). The ultramafic setting of Rainbow is atypical for the region, which is dominated by basalt hosted vent systems (Douville et al., 2002). Due to serpentinization reactions during the circulation of the hydrothermal fluid in the peridotite basement rocks, the Rainbow vent field produced plumes particularly enriched in transition metals (notably Fe, Mn and Cu) and REE (Douville et al., 2002; Findlay et al., 2015). On the contrary the plumes are depleted in hydrogen sulfides (Charlou et al., 1997; Douville et al., 2002), resulting in relatively high metal/sulfide ratios."

8) Introduction, P3, L62: "The same currents will also disperse mining plumes, created in the vicinity of the hydrothermal vent. These mining plumes are therefore likely to interfere with the hydrothermal plume and thus potentially alter baseline (T0) conditions." I don't really understand what you want to say here.

L62 (L68-70) Changed to: "Basic knowledge of natural plumes is essential to be able to discern mining impacts consisting of plumes created in the vicinity of the vent during excavation and by discharge of the return flow which may interfere with the natural hydrothermal plume."

9) Introduction, P3, L64: "...understanding natural plume processes may reveal how ecosystems adapt to elevated turbidity and co-occurring changes in the chemical environment." If you look at the hydrothermal plume as it is at the Rainbow vent, then you will define the close link between the biodiversity and the environmental changes. I don't see how you can address the resilience of plume ecosystem to turbidity changes. I don't get the point. Please clarify.

L64 (L74): Removed the sentence.

We don't want to address the resilience of plume ecosystem to turbidity changes. We want to provide knowledge of hydrothermal plumes in terms of geochemical and microbial community composition.

10) Introduction, P3, L68: "...the composition of the hydrothermal fluid and the associated sediment formed by precipitation from the hydrothermal plume have been established." The sediments are not precipitated from the plume but parts of the polymetallic particles formed within the plume are preserved within the sediment. I don't understand your sentence.

L68 (L76-77): Changed to: "...the composition of the hydrothermal fluid and sediment influenced by fall-out of particulates from the Rainbow and other hydrothermal plumes have been published."

11) Introduction, P3, L70: See the work from Marques et al., 2006

L70 (L79): Added reference to Marques et al. (2006)

12) Introduction, P3, L72: I have done some work on these sediments, especially on REEs. See Chavagnac et al., 2005

L72 (L81): Added Chavagnac et al. (2005) as reference. Changed to "...showed enrichments of Fe, Cu, Mn, V, As and P, as well as REE (Chavagnac et al. (2005), as a result of fallout from the hydrothermal plume."

13) Introduction, P3, L73: "...deposition from the plume is partially being influenced by microbial activity which enhances scavenging and oxidation rates..." I don't understand the link between deposition and enhanced element scavenging by microbial activity. Please rephrase

L72-76 (L81-84): Rephrased to: "It has further been shown that microbial activity influences plume processes (Breier et al., 2012; Dick et al., 2013), such as scavenging and oxidation rates of metals (Cowen and Bruland, 1985; Cowen et al., 1990; Mandernack and Tebo, 1993; Dick et al., 2009),..."

14) Introduction, P3, L76: What are the implications?

L76 (L84): Changed to: "...influencing the local ocean geochemistry."

15) Introduction, P3, L77: Chemiolithoautotrophic? Yes, changed throughout the manuscript.

L77, 78, 405 (L86, L87, 437): "chemolithoautotrophic" L565 (L603): "chemolithoautotrophs"

16) Introduction, P4, L82: See also Borja et al., 2014; Borja et al., 2016; Reed et al., 2015; Orcutt et al., 2011

We have added citations

L79 (L87): Orcutt et al., 2011 L87-88 (L97-99): "Considering the majority of microbial growth is predicted to occur in the neutrally buoyant portion of the plume (Reed et al., 2015), further efforts should be concentrated on sampling this portion of the plume."

17) Introduction, P4, L83: "...dilution of vent associated microorganisms..." I don't understand this part of the sentence. Please clarify.

L83 (L91-92): Changed to "....reduction in dominance of vent associated microorganisms..."

18) Introduction, P4, L84: "...communities associated with the rising plume would disperse with distance from the vent on a scale of metres, showcasing a variable community within the plume." Unclear, please rephrase

L84 (L92-93): Changed to "...suggesting that communities associated with the initial rising plume become diluted on a scale of metres."

19) Introduction, P4, L86: "...dispersed over potentially hundreds of kilometres..." Hydrothermal dissolved iron can be tracked up to 4000 km. See the paper of Resing et al., 2015 The dissolved part can be traced up to 4000 km, however, this is not the case for the particulate part.

Made a change to address this. L86-88 (L95-97): Changed to: "..., remaining traceable in particulate form to at least 50 km away from its source (Severmann et al., 2004), and even up to 4000 km in dissolved form (Resing et al., 2015).

20) Introduction, P4, L90: What do you mean by 'chemical fractionation'?

L90 (L101): Changed "chemical fractionation" to "Geochemical and biological changes".

21) Introduction, P4, P90: "Notably, due to the lack of quantified characteristics of SMS mining plumes (especially the discharge plume), the T0 influence of this hydrothermal plume may act as an analogue for future mining plume impacts." To date, there are no exploitation deep-sea mining sites (soon in the Pacmanus basin). So I don't understand what you want to say by SMS mining plume, and T0 influence. Please rephrase.

L90 (L68-70): Rephrased to: "Basic knowledge of natural plumes is essential to be able to discern mining impacts consisting of plumes created in the vicinity of the vent during excavation and by discharge of the return flow which may interfere with the natural hydrothermal plume."

22) Introduction, P4, L94: "Although it should be kept in mind that discharge plumes will have different physical characteristics as these plumes will have a higher initial density and therefore would tend to sink rather than maintain buoyancy and may have a different release depth." Please provide some references to sustain your text. It is unclear when you refer to natural plume compared to the one generated by deep-sea mining exploitation.

L93 (L70-71): Changed "discharge plumes" to "mining plumes". L94 (L72): Added Gwyther et al., 2008 and Boschen et al., 2013 as references

23) Introduction, P4, L96: Please start with a new paragraph here

L96 (L100): Started new paragraph.

24) Introduction, P4, L97: If you track changes then, you know what are the environmental conditions outside the immediate impact of hydrothermal plume? Is it right?

Yes, in the manuscript we provide comparisons between plume and non-plume influenced waters (i.e. above-plume).

25) Introduction, P4, L100: "By utilising a range of methods that could be useful as monitoring techniques and describing background environments that may be influenced by SMS mining, we contribute to site specific knowledge of the Rainbow hydrothermal vent plume behaviour, associated (trace) metal enrichments and microbial community composition." Too long. Please rephrase. I suspect that you have specific tools for microbial diversity associated to others more specific to chemical monitoring. Is it right?

L97-100 (L101-105): Changed to name the specific tools used for the analyses: "Geochemical and biological changes were studied vertically in the water column and horizontally along the neutrally buoyant plume using HR-ICP mass spectrometry to determine the (trace) metal and REE content of the SPM. Next generation sequencing methods were used to quantify the heterogeneity in the background pelagic system

that was influenced by the hydrothermal plume."

26) Material and methods, P6, L135: I don't understand the term gradient. What do you mean? Please clarify.

L135 (L140): Changed "gradient" to "path".

27) Material and methods, P6, L138: Which type of CTD rosette? Do you follow the GEOTRACES recommendations? Please explain.

Although the method applied by us was similar to the GEOTRACES recommendations, it was not completely similar. Concerning the sampling in general, nutrient samples were taken along with all trace element samples to verify the proper bottle and rosette operation and sampling depth (i.e. to compare the hydrography established with the conventional CTD/Rosette). As recommended by GEOTRACES the filtration was done directly from pressurized bottles and the recommended filters and filter holders were used (Pall Gelman Supur 0.45 $\mu$m polyethersulfone filters and Advantec-MFS 47 mm polypropylene inline filter holders). The filters were acid-cleaned before used. However, our blanks were acid-cleaned unused filters whereas GEOTRACES recommend otherwise to correct for the absorption by the filter.

L138 (L143): CTD was a Seabird 911 system. Changed in text to "Seabird 911 CTD-Rosette system".

28) Material and methods, P6, L140: What do you mean by temporal? This is unclear.

We don't agree as it is mentioned that CTD casts have been taken continuously over 12 hours, to study the temporal changes (i.e. the changes over time).

29) Material and methods, P7, L160: "...or once again if the difference between the two measurements was 0.03 mg or more.". Unclear

L160 (L167-168): Changed to: "...or in triplo if the difference between the first two measurements was more than 0.03 mg."

30) Material and methods, P161: Please provide some additional information about the instrumental procedure you used. Standards?

L161 (L173-174): Added: "The SEM was operated under an acceleration voltage of 15 kV and a filament current of 1850 mA".

31) Results, P10, L250: "Against a background of non-plume influenced waters with typical concentrations of SPM of 0.04 mgL-1 (0.015 NTU)..." Where did you get this information? Is it your data? Or from literature? Please clarify.

L250 (L275): Added: "..., as found in the CTD casts,.." to clarify how these data were obtained.

32) Results, P10, L252: "The apparent continuity of this turbid water layer, especially to the NE of the Rainbow field, and lack of similarly turbid waters in the bottom waters below the plume, link the plume to Rainbow and preclude an origin in local sediment resuspension." This is already the discussion

L379 (L408-410): Moved text above to discussion paragraph 4.1 "The apparent continuity of this turbid water layer, especially to the NE of the Rainbow field, and lack of similarly turbid waters in the bottom waters in the bottoms below the plume, link the plume to Rainbow and preclude local sediment resuspension as origin."

33) Results, P11, L276: The database of geochemical composition is not huge. I wonder whether the statistic treatment is appropriate? Where is the data table?

It is a non-constrained ordination and not a statistical test per se, there are no p-values. It is a visualisation of the similarity between the samples.

A table with the full geochemical dataset (concentrations in pM, with precision in %) will be made public in PANGAEA when the manuscript is published and is also already available in the NIOZ data portal (https://dataverse.nioz.nl/dataverse/doi under DOI 10.25850/nioz/7b.b.s). We have added a table in the supplement (Table S2) showing part of the (trace) metal and REE data as we compare it to other work.

34) Results, P12, L291: Where is the data?

A table with the full geochemical dataset (concentrations in pM, with precision in %) will be made public in PANGAEA when the manuscript is published and is also already available in the NIOZ data portal (https://dataverse.nioz.nl/dataverse/doi under DOI 10.25850/nioz/7b.b.s). We have added a table in the supplement (Table S2) showing part of the (trace) metal and REE data as we compare it to other work.

35) Figure 2, P32: How does it compare to the work of German et al., 1996 in this area? If you want to address the temporal change of hydrothermal plume environment, this is one way to compare the neutrally buoyant plume features 20 years apart. That would be great.

In the discussion we mention the comparison of our results to those of German et al. (1998). L379-381 (L410-413): "Using turbidity measurements and presumed plume path, we traced the plume up to 25 km away from the vent source. This is within the range mentioned by German et al. (1998) who found that the Rainbow plume extends over 50 km, being controlled by local hydrodynamics and topography." Furthermore, we have added a table in the supplement (Table S2), comparing part of our data with o.a. German et al. (1991).

36) Figure 5, P34: It will be interesting to indicate the station? A color coding as in Fig. 6. Did you use the NTU measured at the depth of water collection?

Changed Figure 5 to include the colour coding Changed description of Figure 5 to: "Relationship between in-situ measured turbidity and molar concentration of particulate iron."

37) Figure 6, P35: "Relationship between copper (a), vanadium (b), yttrium (c) and tin (d) to iron" Geochemical analyses of the waters? Or is it the SPM? Not clear. Data?

Changed to "Relationships between molar concentrations of particulate copper (a), vanadium (b), yttrium (c) and iron (d) to iron collected from the filtered water samples.

A table with the full geochemical dataset (concentrations in pM, with precision in %) will be made public in PANGAEA when the manuscript is published and is also already available in the NIOZ data portal (https://dataverse.nioz.nl/dataverse/doi under DOI 10.25850/nioz/7b.b.s). We have added a table in the supplement (Table S2) showing part of the (trace) metal and REE data as we compare it to other work.

38) Figure 7, P36: Comparison with the work of Cave et al., 2002 and Chavagnac et al., 2005, Edmonds and German, 2004

Comparison with work of Cave et al., 2002 and Edmonds and German, 2004 is described in the discussion section 4.3. L473 (L508): Added Chavagnac et al. 2005 as a reference

Please also note the supplement to this comment:
https://www.biogeosciences-discuss.net/bg-2019-189/bg-2019-189-AC1-supplement.pdf

---

## Author Comment (AC2) · 26 Nov 2019

Comments reviewer 2 BG-2019-189

We would like to thank the reviewer for the efforts and input provided, which definitely helped to improve the manuscript. We carefully went through all the comments and suggestions and have adjusted the manuscript according to the comments made. Below we provide descriptions of the adjustments we made, addressing the reviewers remarks.

The responses on the comments are given below, but please note the added supple-

[Figure]

ment where the responses are given with proper formatting.

Note) Line numbers: First original manuscript, second revised manuscript

General comments:

"The link between the different geochemical parameters is not sufficiently detailed. What does the combination of REE and trace metals really bring to the story? Similarly, the link between geochemical parameters and microbial communities is not sufficiently exploited. For example, one of the major results that should have been discussed is Figure S4, which shows the correlations between environmental variables and classes of microorganisms. It is only indicated that there is "a complex array of community drivers within the plume". Moreover, the authors claim that their study represents a T0 before mining activities, but I am not convinced by the analogy between the 2 types of plumes. Indeed, the geochemical characteristics could be similar, the temperature, density, and microbial communities will be totally different.

The aim of this study was to characterize the T0 state of a hydrothermal plume before it is impacted by deep-sea mining to serve as a baseline study which will aid in monitoring of the impacts of plumes created by deep-sea mining, as the situation after mining can then be compared to a state before mining. The plume is characterized in terms of geochemistry and the microbial assemblages as it disperses away from its source. It was not in the scope of this study to exploit the link between the geochemical parameters and microbial communities as we do not have the means to assess all the chemolithoautotrophic and metabolic processes that are going on. The Figure S4 therefore only serves as an initial result and needs to be further studied in future studies. We do agree that our phrasing on an analogue to a mining plume is inappropriate. We have reworded this in the abstract and in the introduction.

Specific comments:

1) Title, P1, L1: I am not convinced that the results show the successional patterns of

trace metals and microorganisms and I would recommend to remove the word "successional".

L1 (L1): Removed "Successional"

2) Material and methods, sampling, P6: The sampling strategy seems confusing to me. Why several stations were sampled at the same location? What is the difference between these stations? The differences observed for the same parameter among the stations are not discussed. SPM, trace metals, and the microbial community are not systematically sampled at the same location. For example, stations 37, 38 and 39 were only sampled for trace metals. Is there any explanation why the different depths of each station were not systematically sampled for all parameters? It is indicated that intermittent water samples were taken for nutrients, but no information is reported on Table 1. For suspended particulate organic matter, I assume the authors refer to C/N on Table 1. No information is given for the analyses of nutrients and POC/PON. I understand that coring sites were constrained by the coring substrate, by why was not CTD deployed at each coring site?

Stations were not sampled at the same location, however they were quite close together to study the small scale variability of the hydrothermal plume, which is why they seem to be at the same spot on the map. The latitude and longitude for each station is added in Table 1.

L146 (L152-155): Added information about sampling: "Depths for sampling SPM were chosen to comprise the largest variation in turbidity measured by the WETLabs turbidity sensor in a vertical profile so that the sensor could be reliably calibrated and readings converted to mgL-1. If possible, trace metal and microbial community samples were taken at the same stations and/or same depth."

We have removed the sentence that additional samples have been taken for nutrients and SPOM as we do not use these samples in our study. The C/N column has also been removed from Table 1.

It is a valid point that no CTD's have been taken at the box core locations. However, as the main focus was to follow the plume along its presumed path no CTD's were taken over the Rainbow Ridge following the box core locations due to time constraints.

3) Material and methods, SPM analyses, P7: I would have liked to see the values of blank filters and the associated uncertainties as well as the average percentage they represent. Please write down what SEM and EDS mean.

Information about the values of the blank and the sampled SPM filters are available at the NIOZ data portal (https://dataverse.nioz.nl/dataverse/doi under DOI 10.25850/nioz/7b.b.s).

L159-L161 (L168-171): Added information about the blanks: "To yield SPM concentrations, the net dry weight of the SPM collected on the filters (average of 0.25 mg), corrected by the average weight change of all blank filters (0.04 mg), was divided by the volume of filtered seawater (5 L)"

L162 (L171-172): Changed "SEM" to "scanning electron microscope (SEM)" and "EDS" to "energy-dispersive spectroscopy (EDS)"

4) Material and methods, P7: This section is missing some important information and is much less detailed than the following one. Were the filters acid-cleaned before use? What are the values for the blank filters? Were procedural blank performed? Which certified reference material was used to assess the accuracy of the analyses?

In L167 (L178) it was stated that the filters were acid-cleaned: "acid-cleaned 0.45 $\mu$m polysulfone filters"

L176 (L188-191): Added information about the procedural blanks: "Furthermore, ten procedural blanks were performed. Half of them were empty acid-cleaned Teflon vials, the other five contained an acid-cleaned blank filter to correct for the dissolved filters. These blanks were subjected to the same total digestion method as described above". Information about the values for the blank filters will be available at the NIOZ data

archive system.

L178 (L193-195): Added information about the calibration: "The concentrations were calculated using external calibration lines made from a multi stock solution, which was prepared by mixing Fluka TraceCert standards for ICP. Rh was used as an internal standard for all elements."

L178 (L195-196): Added information about the drift: "The machine drift was measured before, half-way and after each series of samples and was monitored by using an external drift solution.

L179 (L196-200): Added information about the precision: "Precision (relative standard deviation (RSD)) of these analyses was generally <2 % for major- and trace metals, apart from 115In where the RSD values generally are between 4 % and 8 %, with maximum values going up to 12.48 %. For REE, the RSD values were generally <3 %, apart from a few measurements where RSD values reached maximums up to 12.48 %."

L178 (L200-201): Added information about the accuracy: "The accuracy could not be determined as no certified reference material was analysed."

5) Material and methods, P9: For the biodiversity index, the authors should be consistent along the manuscript. With the name of the index (Shannon-Wiener vs. Shannon).

Changed it to Shannon-Wiener throughout the entire manuscript.  (L342 (L369), change made).

6) Water column characteristics, P10: Using the T-S diagram, the authors identified 3 water masses. However, the hydrography of the area is certainly more complex than that, as shown in the article by Jenkins et al. (2015), even if this later study was located further south

We do agree that the hydrography of the area is more complex, but we wanted to point out the main differences in water masses where we did the sampling.  L240 (L265):

Changed to: "..., whereby three main different water masses could be distinguished."

7) Enrichments of trace metals compared to the ambient seawater, P11: In addition to the enrichments factors, I would have liked to see vertical profiles of the absolute values of trace metals and the range of variations. How was the "clear water" defined?

Clear water is defined as the water above the plume. Changed made in L288 (L313): "clear water above the plume" to "above plume water".

A table with the full geochemical dataset (concentrations in pM, with precision in %) will be made public in PANGAEA when the manuscript is published and is also already available in the NIOZ data portal (https://dataverse.nioz.nl/dataverse/doi under DOI 10.25850/nioz/7b.b.s). We have added a table in the supplement (Table S2) showing part of the (trace) metal and REE data as we compare it to other work.

8) Geochemical gradients, P12: Fe was found to be linearly correlated to the turbidity with a R2 higher than 93%. What was the p value? In the text, it is written that the chalcophile elements Co, Cu and Zn are shown on Fig. 6A, but only Cu is shown. Same for V and P for Fig 6B and REEs for Fig 6C, where only V and Y are shown. Similarly, in the text, Mn, Al, Ni, In, Pb, Ti and U are referred to Fig. 6D, while Sn is shown on this figure.

L297 (L323): "P-value: 2.2*10-16"

Clarified that only one element is shown to illustrate the trend they show. L299 (L326): "Fig. 6A for Cu" L302 (L329): "Fig. 6B for V" L304 (L331): "Fig 6C for Y" L310 (L337): added "Sn" L311 (L338): "Fig. 6D for Sn"

9) L301: the authors state that Zn/Fe ratio is elevated at stations 37, 39 and 44. This is also the case at station 40, and is not discussed in the text.

L301 (L328): Added: "Furthermore, a high Zn/Fe molar ratio is observed at upstream station 40."

10) L302: on Fig 6B the relation between V and Fe indeed looks linear, but the axes are drawn with a logarithmic scale, which means that the relations is not linear but polynomial. The V:Fe ratio is not more or less constant and displays values from 0.005 to $\sim$ 0.0012 (please change also on line 462). It is the same for the REEs.

This is only the case if one of the axes is transformed. If both axes are transformed to a log-scale the same relationships are there as in the case both axes would be on a linear scale. Only if one of the two is on a different axis the relation would be polynomial.

L302 (L329): Changed to: "...and shows varying element/Fe molar ratios without a clear trend of increasing or decreasing ratios". L305 (L333): Removed "constant" L462 (L497): Changed to: "slightly varying"

11) Microbial assemblages, P13, L316 (L343): Please replace "above plume" by "no plume"

Accepted.

12) L317 (L344): Please replace "which clustered distinctively from each other and from plume and below plume communities" by "which clustered distinctly from each other and from plume, below-plume, and above-plume communities"

Accepted.

13) L318 (L345): Please replace "sediment and near-bottom water samples have communities that are very dissimilar from the overlying water column samples" by "sediment, near-bottom water, and no-plume samples have communities that are very dissimilar from the overlying water column samples"

Accepted.

14) Univariate biodiversity, P13: Data used for Fig. 10 and Fig. 11 is slightly confusing. In Fig. 10, the value for diversity index in the plume is about 3.5 with SE lower than 0.5. In Fig. 11, the values for samples in each plume vary from less than 2.5 to higher than

4.5. So I am wondering if the value in Fig. 10 corresponds to the average value of the data in Fig. 11 or not.

The values given are the standard error of the mean and are representative of the values used in figure 11. The only difference is the exclusion of station 13 in figure 10 due to it not being considered a legitimate plume data point.

15) Plume influence on the water column chemical and microbial make-up (P16-17): A table with the range of variation of the literature values would be useful.

The tables are added to the supplement (Table S2).

L400-403 (L432-435): "Our chemical results from Rainbow also match with those of Ludford et al. (1996), who have studied vent fluid samples from TAG, Mid-Atlantic Ridge at Kane (MARK), Lucky Strike and Broken Spur vent sites, i.e. element concentrations were found to be in the same order of magnitude (Table S2)."

16) Line 408 (L440): Please specify here what you mean with oceanic water masses.

We meant the water masses mentioned earlier. Removed the term "oceanic" to avoid any confusion

17) Line 411: Please specify what you mean with SUP05

L411 (L443-444): Added a couple of words to explain that SUP05 is a gammaproteobacteria clade; "...such as the Gammaproteobacteria clade SUP05...".

18) Line 442-443 (L475-477): the authors infer the dependence of sediment dwelling Epsilonproteobacteria on nearby plume precipitates, such as Cu, Zn and Cd, but why only these 3 elements. This should be justified.

Of these elements it is shown that they fall-out of the plume rapidly (both in this study and in others). Added another reference and context to explain this better.

L442-443 (L475-477): "..., thus we infer a relationship between the sediment dwelling

Epsilonproteobacteria with nearby plume precipitates, such as Cu and presumed precipitates Zn and Cd (Trocine and Trefry, 1988)."

19) Geochemical gradients with the hydrothermal plume, P19: The high Ca:Fe ratio at station 40 is explained by the non-influence of hydrothermal plume. Please add a reference for this statement

It is shown in this study that the Ca/Fe ratio is high, as the Fe concentrations are much higher within the hydrothermal plume. Because of this we come up with this statement ourselves. To show another study that shows that the abundance of particulate iron is low in water which aren't influenced by the hydrothermal plume Michard et al. (1984) is added an a reference.

L483-486 (L519-523): "The high molar ratio at staion 40 would then suggest that this station is hardly or not at all influenced by the hydrothermal plume as the natural abundance of particulate iron is low (e.g. Michard et al., 1984 and this study), whereas station 28, 47 and 49 are, as expected, influenced in more moderate degrees compared with the station directly downstream of Rainbow."

20) Microbial gradients within the hydrothermal plume, P20: The authors state that the dominance of Epsilonproteobacteria is likely driven by the strong chemical enrichment of the plume but when looking at Fig. S4, Epsilonproteobacteria is not within the group that is most strongly positively correlated with trace metals. As I wrote above, this point would be very interesting to discuss as well as the other correlations.

Looking into such patterns required much more rigorous statistical testing, something we cannot do with the number of samples we have. Furthermore, we are reluctant to correlate continuous data with proportional data (microorganisms) with full confidence of inferring relatable patterns. Added information in the introduction to better emphasise the aim of this study: L103 (L105-109): "Whilst mechanic understanding of microbial and geochemical interactions in the plume would have required a different experimental setup, which was beyond the scope of the TREASURE project, this

paper aims to contribute to knowledge of geochemical and biological heterogeneity in the surrounding of an SMS site, induced by the presence of an active hydrothermal plume, which should be taken into account in environmental impact assessments of SMS mining."

21) L511-513 (L549-551): This statement is too speculative

L511-513 (L549-550): Altered the language, changed to "These patterns may relate to ecological succession (Connell and Slaytor, 1977) within the plume. . ."

L513-515 (L551-553): The use of likely probably created a too speculative tone, therefore we changed from "likely" to "possibly". No other hypotheses are put forward.

Figures and tables:

22) Fig. 1: Station 30 is indicated twice.

Changed one 30 to 33.

23) Fig. 2: The x axis represents the distance from Rainbow. On Fig. 1 it looks like station 44 is located closer to Rainbow than station 26

That's because we measured the distances to Rainbow along the transect of the plume instead of its direct distances. Changed the description of Fig. 2 to include that it follows the plume transect as found in Fig. 1 "Transect along main plume path (indicated in Fig. 1 as plume transect), showing turbidity in the water column. The plume is indicated by highest turbidity values and disperses away from the Rainbow vent field."

24) Table 1: Could you indicate long-lat for each station?

Added latitude and longitude for the stations.

Please also note the supplement to this comment: https://www.biogeosciences-discuss.net/bg-2019-189/bg-2019-189-AC2-supplement.pdf

---

## Author Response (AR2)

**Report reviewer 2 BG-2019-189**

We would like to thank the reviewer for reviewing the revised manuscript and the input provided for the additional corrections. We carefully went through these comments and have adjusted the manuscript according the comments made. Below we provide descriptions of the adjustments we made, addressing the reviewers remarks.

1) **P1, Line 19: Please abbreviate "rare earth elements" as REEs and not REE and correct in the manuscript where applicable (e.g. lines 31, 49, ….)**

   Changed REE to REEs in the following lines: 19, 33, 38, 50, 82, 125, 201, 254, 336, 435, 638, 892.

2) **P2, Line 28: Please replace "HR-ICP mass spectrometry" by "Sector Field Inductively Coupled Plasma Mass Spectrometer (SF-ICP-MS) with high resolution (HR)" and in the rest of the following (lines 102, 191 and 628).**

   We have changed "HR-ICP mass spectrometry" by "Sector field inductively coupled plasma mass spectrometry (SF-ICP-MS) with high resolution (HR)" and changed the "HR-ICP-MS" abbreviation to "SF-ICP-MS" throughout the text.

   L28-29: "HR-ICP mass spectrometry" was changed to "sector field inductively coupled plasma (SF-ICP-MS) with high resolution (HR)".

   L103-104: "HR-ICP mass spectrometry" was changed to "sector field inductively coupled plasma (SF-ICP-MS) with high resolution (HR)".

   L193: "HR-ICP-MS" was changed to "SF-ICP-MS".

   L639: "HR-ICP-MS" was changed to "SF-ICP-MS".

3) **P6, Line 145: Salinity is without unit. Please remove "PSU"**

   Removed "PSU" throughout the manuscript. Changes made at lines: 147, 272, 274, 276.

4) **P8: For blank determination, the authors now give more detail. Values will be available on the NIOZ database but I suggest that the authors give some range here or percentages of the measured values**

[revised manuscript text omitted]